# ARIH1 signaling promotes anti-tumor immunity by targeting PD-L1 for proteasomal degradation

Youqian Wu[1,11], Chao Zhang[2,3,11], Xiaolan Liu[1,11], Zhengfu He[4,11], Bing Shan[5], Qingxin Zeng[4], Qingwei Zhao[1], Huaying Zhu[6], Hongwei Liao[7], Xufeng Cen[1], Xiaoyan Xu[1], Mengmeng Zhang[5], Tingjun Hou [8], Zhe Wang[8], Huanhuan Yan[1], Shuying Yang[1], Yaqin Sun[1], Yanying Chen[1], Ronghai Wu[1], Tingxue Xie[1], Wei Chen [6], Ayaz Najafov [9,12✉], Songmin Ying [2,7,12✉] & Hongguang Xia [1,10,12✉]

Cancer expression of PD-L1 suppresses anti-tumor immunity. PD-L1 has emerged as a remarkable therapeutic target. However, the regulation of PD-L1 degradation is not understood. Here, we identify several compounds as inducers of PD-L1 degradation using a high-throughput drug screen. We find EGFR inhibitors promote PD-L1 ubiquitination and proteasomal degradation following GSK3α-mediated phosphorylation of Ser279/Ser283. We identify ARIH1 as the E3 ubiquitin ligase responsible for targeting PD-L1 to degradation. Overexpression of ARIH1 suppresses tumor growth and promotes cytotoxic T cell activation in wild-type, but not in immunocompromised mice, highlighting the role of ARIH1 in anti-tumor immunity. Moreover, combining EGFR inhibitor ES-072 with anti-CTLA4 immunotherapy results in an additive effect on both tumor growth and cytotoxic T cell activation. Our results delineate a mechanism of PD-L1 degradation and cancer escape from immunity via EGFR-GSK3α-ARIH1 signaling and suggest GSK3α and ARIH1 might be potential drug targets to boost anti-tumor immunity and enhance immunotherapies.

---

[1] Department of Biochemistry and Research Center of Clinical Pharmacy of The First Affiliated Hospital, Zhejiang University School of Medicine, Hangzhou 310058, China. [2] International Institutes of Medicine, The Fourth Affiliated Hospital of Zhejiang University School of Medicine, Yiwu 322000, China. [3] Key Laboratory Respiratory Disease of Zhejiang Province, Department of Anatomy and Department of Respiratory and Critical Care Medicine of the Second Affiliated Hospital, Zhejiang University School of Medicine, Hangzhou, Zhejiang 310009, China. [4] Department of Thoracic Surgery, Sir Run Run Shaw Hospital, School of Medicine, Zhejiang University, Hangzhou, Zhejiang 310016, China. [5] Interdisciplinary Research Center on Biology and Chemistry, Shanghai Institute of Organic Chemistry, Chinese Academy of Sciences, Shanghai 201203, China. [6] Department of Cell Biology and Department of Cardiology of the Second Affiliated Hospital, Zhejiang University School of Medicine, Hangzhou 310058, China. [7] Key Laboratory Respiratory Disease of Zhejiang Province, Department of Pharmacology and Department of Respiratory and Critical Care Medicine of the Second Affiliated Hospital, Zhejiang University School of Medicine, Hangzhou, Zhejiang 310009, China. [8] Hangzhou Institute of Innovative Medicine, College of Pharmaceutical Sciences, Zhejiang University, Hangzhou 310058, China. [9] Department of Cell Biology, Harvard Medical School, Boston, MA 02115, USA. [10] Liangzhu Laboratory, Zhejiang University Medical Center, Hangzhou 311121, China. [11]These authors contributed equally: Youqian Wu, Chao Zhang, Xiaolan Liu, Zhengfu He. [12]These authors jointly supervised this work: Ayaz Najafov, Songmin Ying, Hongguang Xia. ✉email: ayaz_najafov@hms.harvard.edu; yings@zju.edu.cn; hongguangxia@zju.edu.cn

Programmed death ligand-1 (PD-L1) is constitutively expressed on the surface of cancer cells[1–4]. The interaction between PD-L1 and its receptor, programmed death protein-1 (PD-1), which is mainly expressed on the surface of T cells, results in cancer cell evasion from immune surveillance[5]. PD-1 and PD-L1 have become important immune checkpoint blockade immunotherapy targets for different types of cancers[5,6]. However, resistance to such immunotherapy approaches is prevalent and the mechanisms of resistance are not well understood.

Proteasomal degradation of PD-L1 has been reported to be promoted by cyclin D-CDK4-mediated phosphorylation followed by Cullin 3$^{SPOP}$-dependent ubiquitination[7], as well as glycogen synthase kinase 3-β (GSK3β)-mediated phosphorylation followed by β-TrCP-dependent ubiquitination[8]. However, the dynamic ubiquitin modification of PD-L1 for proteasomal degradation should mostly occur in the intracellular segment of PD-L1, which is not addressed before. Ariadne-1 homolog (ARIH1) is a member of the Ariadne family of E3 ubiquitin ligases with a cognate E2 enzyme UBCH7[9–11]. ARIH1 is known to play a role in protein translation regulation in response to DNA damage and to ubiquitinate EIF4E2[12]. The role of ARIH1 in PD-L1 degradation or anti-tumor immunity is not known.

Epidermal growth factor receptor (EGFR) is a major receptor tyrosine kinase, frequently overactivated in cancers, with established roles in cell growth and survival[13–15]. Patients with EGFR-mutant-driven tumors develop resistance to EGFR tyrosine kinase inhibitor treatments, which are usually accompanied by the acquisition of EGFR mutations[16]. Clinical retrospectives suggest that EGFR mutations are associated with low response rates to immune therapies in non-small cell lung cancer[17,18]. Importantly, EGFR mutations are associated with increased PD-L1 expression[19–21]. EGFR activation promotes PD-L1 expression via Janus kinase/signal transducer and activator of transcription 3 (JAK/STAT3) signaling pathway[22] and inhibits its degradation via phosphorylation mediated by GSK3β[8] in the extracellular part of PD-L1, contributing to cancer escape from anti-tumor immunity. These findings strongly indicate that EGFR works as a critical regulator of tumor immune surveillance via regulation of PD-L1 expression, but the mechanism of this regulation is not fully understood, especially as the reported phosphorylation sites are in the extracellular part of PD-L1.

In this study, we screened a panel of 2125 Food and Drug Administration (FDA)-approved drugs or drug candidates and found that ES-072, a third-generation EGFR inhibitor, induced a potent degradation of PD-L1. Our mechanistic findings suggest that inhibition of EGFR activates GSK3α by suppressing AKT activity, which subsequently promotes the phosphorylation at the intracellular Ser279 and Ser283 residues of PD-L1, leading to ARIH1-mediated ubiquitination and proteasome-mediated degradation. Cancer-associated mutation of ARIH1 compromises ubiquitination of PD-L1. Together, our results suggest ARIH1 is the E3 ligase for PD-L1, which could lead to the development of therapeutic strategies to overcome immunotherapy resistance in cancers and enhance checkpoint blockade therapy efficacy.

## Results

**A high-throughput screen of 2125 FDA-approved drugs or drug candidates identifies promoters of PD-L1 degradation.** To establish a fluorescence-based high-throughput screening protocol for the membrane levels of endogenous PD-L1, we used a phycoerythrin-conjugated anti-PD-L1 antibody and interferon-γ (IFNγ)-treated U937 cells (histiocytic lymphosarcoma cell line). IFNγ enhances the basal expression levels of PD-L1, allowing for a wider dynamic range and, thus, a better screening system[23,24].

As a positive control for drug-induced PD-L1 level decrease, we used Ruxolitinib, a JAK1/2 inhibitor[25] (Supplementary Fig. 1a). As expected, PD-L1 levels were induced by IFNγ and blocked by Ruxolitinib treatment (Supplementary Fig. 1b–d).

Out of 2125 FDA-approved drugs or drug candidates screened, 160 were found to reduce the membranal PD-L1 levels (Supplementary Fig. 1e and Supplementary Table 1). We classified the positive hits according to the signaling pathways they were known to be involved in, which included the JAK/STAT pathway, the phosphatidylinositol 3-kinase (PI3K)/AKT/mammalian target of the rapamycin pathway, pathways that regulate the cell cycle, and protein tyrosine kinases (Fig. 1a). The screen confirmed the existing knowledge about PD-L1 regulation, as 13 different JAK inhibitors were found to decrease plasma membrane PD-L1 levels by more than 25%; this also indicated that the screen setup is valid, as JAK was the top drug target inhibition of which resulted in the PD-L1 level decrease.

On the other hand, EGFR has also been strongly linked to PD-L1 protein level regulation. From the EGFR inhibitors that promote PD-L1 level decrease, AZD9291 and ES-072 were selected for our screen follow-up and mechanistic studies to investigate what drives PD-L1 degradation downstream of EGFR inhibition. AZD9291, Osimertinib, is a potent and selective mutant EGFR (L858R/T790M) inhibitor, which is widely used in clinic[26,27]. ES-072 is a third-generation EGFR inhibitor designed by our team to overcome drug-resistance-induced EGFR mutation L858R/T790M. Confirming the screen findings, both ES-072 and AZD9291 dramatically reduced the IFNγ-induced membranal PD-L1 levels in U937 cells following treatment with the compounds, as judged by western blotting and flow cytometry results (Fig. 1b–d). The effect of the EGFR inhibitors on PD-L1 levels was time-dependent, reaching a plateau following 12 h of treatment (Fig. 1e–g). Similar findings were seen in H1975 cells (Supplementary Fig. 2a–d) and interleukin-4 (IL-4)-treated peritoneal-derived macrophages (PDMs) (Supplementary Fig. 2e–g).

EGFR inhibitors are known to promote PD-L1 degradation via the GSK3α/β-TrCP pathway[8]. Surprisingly, contrary to the previous reports, we found that knockdown of β-TrCP did not fully block the reduction of PD-L1 levels induced following EGFR inhibition (Fig. 1h, i), whereas GSK3α/β inhibitor LY2090314 did block this reduction (Fig. 1j–l). LY2090314 was found to be highly specific for GSK3α with an half maximal inhibitory concentration (IC50) value of 0.87 ± 0.09 nM (Supplementary Table 2). These results indicated that a β-TrCP-independent, yet GSK3α/β-dependent mechanism regulating PD-L1 levels downstream of EGFR should exist.

**ES-072 promotes proteasomal PD-L1 degradation via EGFR inhibition.** The rest of our studies were continued using ES-072 due to its stability and specificity (Supplementary Table 3). As, upon EGFR inhibition, PD-L1 has been shown to be degraded via the proteasome, we first determined whether ES-072 treatment also results in a proteasome-dependent degradation of PD-L1. Both western blotting and flow cytometry analysis of plasma membrane PD-L1 levels confirmed that EGFR inhibition results in a proteasome-mediated PD-L1 degradation, as the decrease in the PD-L1 levels induced by ES-072 was rescued by the proteasome inhibitor MG132 (Supplementary Fig. 3a–c).

In accord with previous reports[28,29], EGF treatment promoted an increase of PD-L1 levels (Supplementary Fig. 3d–f), whereas small interfering RNA (siRNA)-mediated knockdown of EGFR decreased PD-L1 levels (Supplementary Fig. 3g). ES-072 was found to be highly specific for EGFR in an in vitro kinase-profiling screen, where it was found to inhibit wild-type,

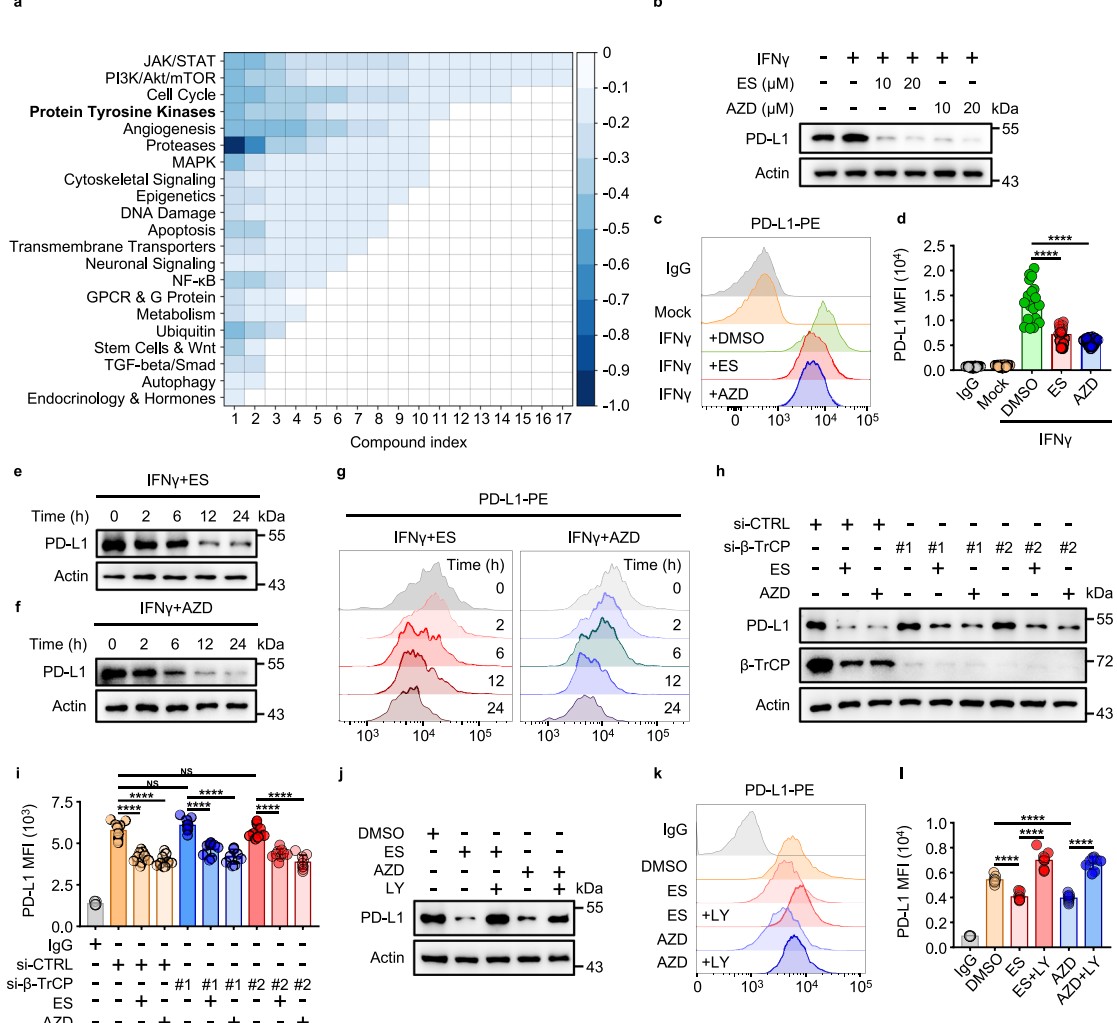

**Fig. 1 EGFR inhibitors were screened to reduce membrane PD-L1 levels. a** High-throughput screening of 2125 FDA-approved drugs or drug candidates. U937 cells were incubated with IFNγ (100 ng/mL) for 48 h, treated with the drugs at 10 μM for 12 h. Ruxolitinib (Rux) was used as a positive control. The hit compounds that induced the decrease of PD-L1 levels are shown in blue. The depth of blue represents decreased level of PD-L1. The heatmap represents the targeted pathways obtained from the high-throughput screening, based upon decreased membrane PD-L1 level detected by flow cytometry. **b** Immunoblotting of PD-L1 in U937 cells treated with ES-072 (ES) or AZD9291 (AZD) at indicated concentrations for 12 h. ES and AZD are EGFR inhibitors. **c, d** Median fluorescence intensity (MFI) (**c**) and relative quantification (**d**) of PD-L1 in U937 cells treated with 10 μM ES-072 or 10 μM AZD9291 for 12 h. Data represent means ± SEM, $n = 18, 6$ independent repeats, ****$P < 0.0001$. **e–g** Immunoblotting (**e, f**) and flow cytometry (**g**) analysis of PD-L1 levels in U937 cells treated with 10 μM ES-072 or 10 μM AZD9291 for the indicated times. **h, i** Immunoblottings (**h**) of PD-L1 and β-TrCP, relative quantification (**i**) of PD-L1 MFI in H1975 cells transfected with non-targeting siRNA (si-CTRL) or β-TrCP siRNAs (si-β-TrCP) and treated with or without 10 μM ES-072/AZD9291 for 12 h. Data represent means ± SEM, $n = 9$, 3 independent repeats, NS: no significant; ****$P < 0.0001$. **j–l** Immunoblotting (**j**) and flow cytometry analysis (**k**) with relative quantification (**l**) of PD-L1 in H1975 cells treated with 10 μM ES-072 or AZD9291, and/or 5 μM LY2090314 (LY) for 12 h. LY is a GSK3 inhibitor. Data represent means ± SEM, $n = 9$, 3 independent repeats, ****$P < 0.0001$. Source data are provided as a Source Data file.

T790M, and T790M/L858R EGFR with IC50 values of 8.9, <0.5, and 1.75 nM, respectively (Supplementary Table 3). Furthermore, the decrease of PD-L1 levels induced by ES-072 is dependent on EGFR (Supplementary Fig. 3h). These results suggest that the effect of ES-072 on PD-L1 is via EGFR, consistent with the notion that ES-072 treatment targets PD-L1 for proteasomal degradation, whereas EGF treatment decreased the basal level of these ubiquitination events (Supplementary Fig. 3i). Together, these data indicated that ES-072 induced the decrease of PD-L1 through inhibition of EGFR, induction of K48-linked ubiquitination, and proteasomal degradation.

**PD-L1 degradation depends on its phosphorylation at Ser279 and Ser283.** To determine the mechanism behind ES-072-induced

PD-L1 degradation, we tested whether PD-L1 undergoes any phosphorylation changes following treatment with this EGFR inhibitor. As judged by mass spectrometry, ES-072 induced several PD-L1 phosphorylation events (Fig. 2a–c and Supplementary Table 4). The top two most robustly induced phosphorylation events on the cytosolic side of PD-L1 were on Ser279 and Ser283 (Fig. 2a, b and Supplementary Table 4). It should be noted that both Ser279 and Ser283 are located in the intracellular segment of PD-L1 (Fig. 2c). Mutating these residues to the phosphorylation-resistant alanine showed that both of the sites play an important role in ES-072-induced PD-L1 degradation, and that dual S279A; S283A (2SA) mutation blocks PD-L1 degradation more potently than either S279A or S283A alone (Fig. 2d). Consistently, chase experiments using protein synthesis inhibitor cycloheximide showed that these

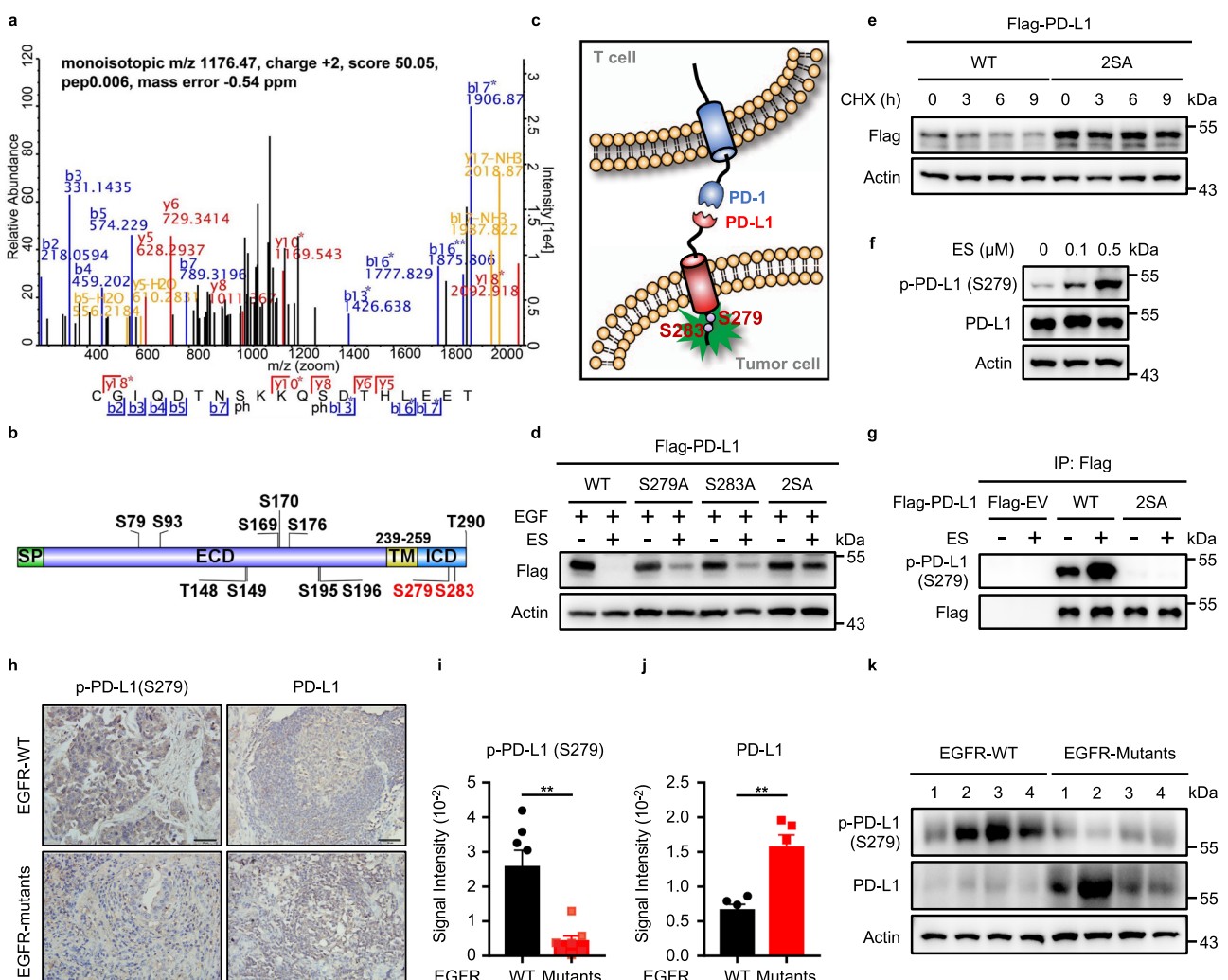

**Fig. 2 Phosphorylation on S279/S283 impairs the stability of PD-L1. a** Mapping PD-L1 phosphorylation sites following ES-072 treatment. HEK293T cells were transfected with Flag-PD-L1 and treated with 10 μM ES-072 for 48 h. Immunoprecipitated PD-L1 was analyzed by mass spectrometry following phosphopeptide enrichment. Peptide ionization data corresponding to Ser279/283 are shown. **b** Schematic diagram of phosphorylated sites on PD-L1. Full-length PD-L1 was separated into an extracellular domain (ECD) and intracellular domain (ICD). SP: signal peptide; TM: transmembrane domain. **c** Schematic diagram of phosphorylated sites, S279 and S283 in the ICD of PD-L1. Positions of phosphorylated sites were labeled in red. This image was created by the first author. **d** Immunoblots of PD-L1 (anti-Flag) in HEK293T cells transfected with Flag-PD-L1 (WT) or Flag-PD-L1 (mutants) following treatment with 25 ng/mL EGF and/or 10 μM ES-072 for 48 h, 2SA represents S279A/S283A. **e** Immunoblots of PD-L1 (anti-Flag) in HEK293T cells transfected with Flag-PD-L1 (WT/2SA) following treatment with 20 μg/mL cycloheximide (CHX) for indicated times. **f** Immunoblots of H1975 cell lysates following ES-072 treatment for 2 h, at indicated doses. **g** HEK293T cells were transfected with Flag-PD-L1 (WT/2SA) following treatment with 10 μM ES-072 for 2 h; PD-L1 was immunoprecipitated with anti-Flag and immunoblotted with indicated antibodies. Flag-tagged empty vector (Flag-EV) was transfected as a negative control. **h** Representative images of p-PD-L1 (Ser279) and PD-L1 immunohistochemistry (IHC) staining from EGFR wild-type vs. mutant human alveolar adenocarcinoma specimens. Scale bars represent 50 μm. **i, j** Quantification of IHC analysis for p-PD-L1 (Ser279) (**i**) and PD-L1 (**j**) in **h**. Data represent means ± SEM, $n = 7$ (**i**), $n = 5$ (**j**), **$P < 0.01$, $P = 0.0013$ (**i**); $P = 0.002$ (**j**). **k** Immunoblottings of p-PD-L1 (Ser279) and PD-L1 in EGFR wild-type ($n = 4$) vs. mutant ($n = 4$) human lung adenocarcinoma specimens. Source data are provided as a Source Data file.

phosphorylation events are critical for PD-L1 turnover, as the 2SA mutation delayed PD-L1 degradation (Fig. 2e).

To confirm that endogenous PD-L1 is phosphorylated following ES-072 treatment, we generated a phospho-specific antibody against PD-L1 Ser279. Consistent with mass spectrometry data, ES-072 treatment induced robust phosphorylation at Ser279 in both endogenous and recombinant PD-L1, which was blocked by the S279A mutation, indicating the specificity of the antibody (Fig. 2f, g). Notably, immunohistochemistry and western blotting analysis of tumor biopsies obtained from alveolar adenocarcinoma patients showed that PD-L1 phosphorylation levels at Ser279 are higher in EGFR-WT tumors compared to EGFR-mutant-driven tumors, and this difference in

phosphorylation levels correlated with lower PD-L1 levels in EGFR-WT tumors (Fig. 2h–k). The EGFR mutations found in these patients were L858R, E542K, P753R, and Δex19, which have been linked to the activation of EGFR and the resistance to EGFR inhibitors in non-small cell lung cancer[30].

These results indicate that PD-L1 phosphorylation events at Ser279 and Ser283 are regulated downstream of the EGFR signaling pathway and are critical for PD-L1 degradation.

**EGFR inhibition-induced PD-L1 phosphorylation at Ser279/283 is mediated by GSK3α.** GSK3β activation has been previously shown to promote the degradation of PD-L1 downstream

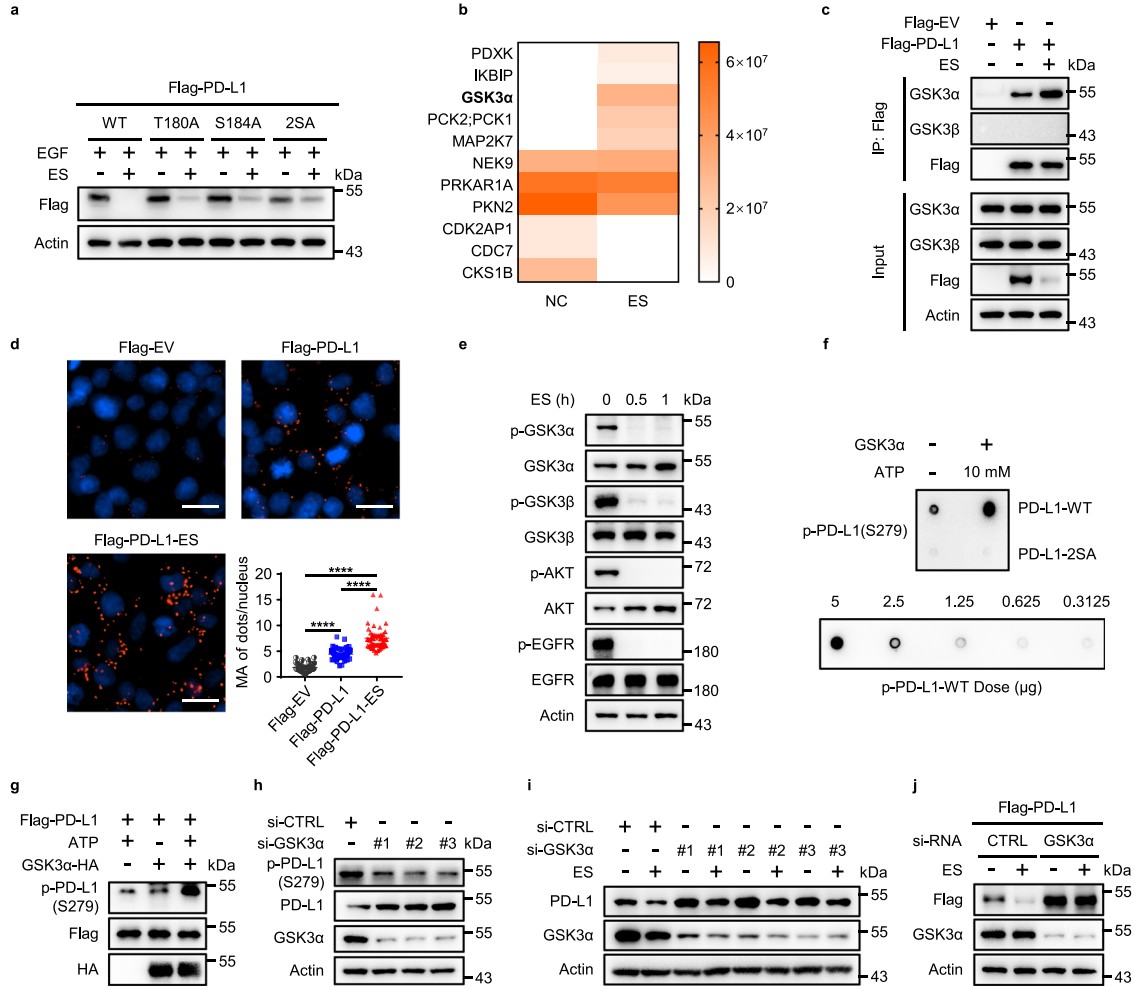

**Fig. 3 GSK3α-mediated phosphorylation of PD-L1 promotes PD-L1 degradation. a** Immunoblots of PD-L1 (anti-Flag) in HEK293T cells transfected with Flag-PD-L1 (WT) or Flag-PD-L1 (mutants) following treatment with 25 ng/mL EGF and/or 10 μM ES-072 for 48 h, 2SA represents T180A/S184A. **b** HEK293T cells were transfected with Flag-PD-L1 and treated with ES-072 (10 μM) for 48 h. Proteins that co-immunoprecipitated (Co-IP) with PD-L1 were analyzed by mass spectrometry. Kinases and kinase-related proteins are shown in heatmap. **c** Co-IP analysis for the interaction of GSK3α/GSK3β and PD-L1 in HEK293T cells transfected with Flag-PD-L1 and treated with or without 10 μM ES-072 for 12 h, Flag-tagged empty vector (Flag-EV) was transfected as a negative control. **d** Proximity ligation assay (PLA) analysis for the interaction of GSK3α and PD-L1 in HEK293T cells treated as **c**. PLA signals are shown in red and the nuclei in blue; scale bar, 20 μm. Quantification for the mean area (MA) of PD-L1/GSK3α PLA speckles is indicated by scattergram. Data represent means ± SEM, $n = 50$, ****$P < 0.0001$. **e** Immunoblots of p-GSK3α, GSK3α, p-GSK3β, GSK3β, p-AKT, AKT, p-EGFR, and EGFR in H1975 cells treated with 10 μM ES-072 for indicated times. **f** In vitro GSK3α kinase assay was performed in the presence or absence of ATP. The phosphorylation of PD-L1 peptides (PD-L1-WT/2SA) was detected by dot blot with anti-p-PD-L1 (Ser279) antibody, p-PD-L1 peptides were synthesized as a positive control. **g** In vitro GSK3α kinase assay was performed in HEK293T cells transfected with Flag-PD-L1 and GSK3α-HA. Total cell lysates were immunoprecipitated with anti-Flag or anti-HA. The phosphorylation of PD-L1 by GSK3α was detected using an anti-p-PD-L1 (Ser279) antibody. **h** Immunoblots of p-PD-L1, PD-L1, and GSK3α in H1975 cells transfected with GSK3α-siRNAs. **i** Immunoblots of PD-L1 and GSK3α in U937 cells transfected with GSK3α-siRNAs and treated with or without 10 μM ES-072 for 24 h. **j** Immunoblots of PD-L1 (anti-Flag) and GSK3α in HEK293T cells transfected with Flag-PD-L1 following treatment with GSK3α-siRNA and treated with or without 10 μM ES-072 for 24 h. Source data are provided as a Source Data file.

of EGFR inhibition, because the loss of EGFR activity results in a decrease of AKT activity, which normally activates both GSK3α and GSK3β[31]. We found that mutating the previously reported GSK3β sites on PD-L1 (T180A and S184A) did not fully block ES-072-induced PD-L1 degradation (Fig. 3a), indicating that ES-072 promotes PD-L1 degradation independent of the previously reported GSK3β-driven mechanism.

To determine the kinase responsible for the Ser279/283 phosphorylation events that drive PD-L1 degradation following ES-072 treatment, we performed a large-scale PD-L1 immuno-precipitation experiment followed by mass spectrometry-based proteomic analysis, which revealed several interactors of PD-L1 (Fig. 3b). These results were also confirmed by western blotting experiments (Fig. 3c). Notably, the top endogenous kinase, the

interaction of which with PD-L1 was enhanced by ES-072 treatment, was found to be GSK3α, but not GSK3β, despite equal expression levels for GSK3α and GSK3β in HEK293T cells (Fig. 3b, c). This result was also confirmed by tunicamycin treatment, a specific N-linked glycosylation inhibitor, using HEK293T cells with exogenous PD-L1 expression (Supplementary Fig. 4a–c).

We further validated these findings by testing for the ability of ES-072 to induce the interaction between PD-L1 and endogenous GSK3α, using proximity ligation assay (PLA), which showed that there is a basal level of GSK3α/PD-L1 interaction, which is enhanced upon ES-072 treatment (Fig. 3d). Interestingly, Ser279 and Ser283 are also a potential match for the consensus motif of GSK3 (p[S/T]XXXp[S/T]p) (Supplementary Fig. 4d)[32,33]. ES-072

treatment promoted GSK3α activation in an EGFR- and AKT-dependent manner, as judged by the decrease of phospho-GSK3α at Ser21, which is a marker of GSK3α inhibition (Fig. 3e). ES-072 showed a higher activation effect on GSK3α when compared to the other two EGFR inhibitors, as judged by loss of the inhibitory phosphorylation of GSK3α (Supplementary Fig. 4e). These findings suggest that GSK3α could be the kinase that directly phosphorylates PD-L1 at Ser279/283 following EGFR inhibition by ES-072.

Consistent with this hypothesis, purified GSK3α robustly phosphorylated wild-type, but not 2SA synthetic peptides in vitro, at Ser279, which we used to represent the intracellular region of PD-L1 (Fig. 3f). Furthermore, purified GSK3α also robustly phosphorylated purified PD-L1 in vitro, at Ser279 (Fig. 3g). Consistently, knockdown of GSK3α, but not GSK3β, blocked the phosphorylation of PD-L1 at Ser279 (Fig. 3h and Supplementary Fig. 4f). These results indicate that GSK3α directly phosphorylates PD-L1 at Ser279. In agreement with the data showing that Ser279/Ser283 phosphorylation is important for ES-072-induced PD-L1 degradation, and that these sites are phosphorylated by GSK3α, knockdown of GSK3α in U937 and H1975 cells partially rescued PD-L1 degradation induced by ES-072 (Fig. 3i and Supplementary Fig. 4g). Similar results were seen when GSK3α was knocked down in HEK293T cells overexpressed with Flag-PD-L1 (Fig. 3j).

Overall, these findings showed that ES-072 treatment results in inhibition of EGFR and activation of GSK3α, which phosphorylates PD-L1 at its cytosolic Ser279 and Ser283 residues to target PD-L1 for proteasomal degradation. Our results also indicate that this mechanism is distinct from the previously reported phosphorylation of PD-L1 by GSK3β at T180 and S184, at the extracellular region of PD-L1[8].

**E3 ubiquitin ligase ARIH1 directly ubiquitinates PD-L1 and targets it for proteasomal degradation, following EGFR inhibition.** To determine which E3 ubiquitin ligase promotes PD-L1 ubiquitination and proteasomal degradation following ES-072 treatment, we analyzed our mass spectrometry immunoprecipitation data for the presence of known mediators of ubiquitination. In addition to cullin ligases, which are known to mediate PD-L1 degradation[7,8], we found that ES-072 promoted PD-L1 interaction with E3 ubiquitin ligases called ARIH1 and ring finger protein 25 (RNF25)[34] (Fig. 4a).

We found that transient overexpression of ARIH1 (ARIH1-OE), but not that of RNF25, strongly promotes K48-linked ubiquitination of Flag-PD-L1 (Supplementary Fig. 5a). Hemagglutinin (HA)-tagged ARIH1 interacted with endogenous PD-L1 (Fig. 4b) and purified GST-PD-L1 could pull down exogenous ARIH1 from HEK293T cell lysates (Fig. 4c). Importantly, ES-072 treatment enhanced the interaction between ARIH1 and PD-L1 (Fig. 4d), consistent with the mass spectrometry data shown in Fig. 4a. In agreement with the notion that ARIH1 promotes PD-L1 degradation, the turnover of PD-L1 in cycloheximide-chased HEK293T cells was exacerbated by ARIH1-OE (Fig. 4e) and was inhibited by knockdown of ARIH1 (Fig. 4f).

Moreover, ARIH1 overexpression in H1975 and HEK293T cells dose-dependently promoted endogenous PD-L1 degradation (Fig. 4g and Supplementary Fig. 5b), whereas ARIH1 knockdown resulted in accumulation of endogenous PD-L1 levels (Fig. 4h and Supplementary Fig. 5c). The ARIH1-induced PD-L1 ubiquitination was blocked by the C357S ligase-dead mutation of ARIH1[9], confirming the involvement of the catalytic activity of this E3 ubiquitin ligase (Supplementary Fig. 5d). Consistently, ARIH1 knockdown reduced basal K48-linked PD-L1 ubiquitination, whereas ARIH1 overexpression dramatically increased it

(Fig. 4i). Importantly, ES-072-induced PD-L1 degradation and K48-linked ubiquitination were rescued by ARIH1 knockdown (Fig. 4j, k). Mutational analysis revealed that ARIH1-driven PD-L1 ubiquitination could be blocked by Lys/Arg point mutations of K271 and K281, but not that of K263, K270, or K280 (Supplementary Fig. 6a, b). Importantly, purified ARIH1 ubiquitinated purified PD-L1 in vitro and this was blocked by C357S ligase-dead mutation of ARIH1 (Fig. 4l and Supplementary Fig. 5e).

These results indicate that ARIH1 directly ubiquitinates PD-L1 and targets it for proteasomal degradation, following ES-072-induced inhibition of EGFR.

**Phosphorylation of PD-L1 at Ser279/283 mediated by GSK3α promotes PD-L1/ARIH1 interaction and subsequent PD-L1 ubiquitination and degradation.** We next tested whether GSK3α-mediated PD-L1 phosphorylation at Ser279/283 is the mechanism of recruitment of ARIH1 to PD-L1. HEK293T cells stably expressing ARIH1-HA were transfected with wild-type PD-L1 or the phosphorylation-resistant S279A, S283A, and 2SA mutants. Remarkably, PD-L1 ubiquitination induced by ARIH1 overexpression was partially reduced by the S279A and S283A mutations, and strongly inhibited by the 2SA double mutation of these GSK3α phosphorylation sites (Fig. 5a). In accord with this, ARIH1 interaction with PD-L1 was strongly reduced by the 2SA mutation (Fig. 5a, HA blot in the IP-Flag-PD-L1 panel).

Importantly, ES-072 treatment enhanced the interaction between ARIH1 and wild-type PD-L1 WT, but not that of PD-L1 2SA mutant (Fig. 5b), whereas overexpression of GSK3α induced PD-L1 ubiquitination, which was blocked by ARIH1 knockdown (Fig. 5c). Notably, GSK3α inhibitor LY2090314 rescued the ARIH1-overexpression-induced degradation of PD-L1 (Fig. 5d), as well as reduction of its membrane levels (Fig. 5e, f) and K48-linked ubiquitination (Fig. 5g).

These findings suggest that PD-L1 degradation is mediated by its GSK3α-driven phosphorylation at Ser279/283 and subsequent recruitment of ARIH1 to ubiquitinate PD-L1 via K48-linked ubiquitin chains that target it for proteasomal degradation.

**ARIH1 promotes anti-tumor immunity via PD-L1 degradation.** Immunohistochemistry analysis of biopsies obtained from healthy volunteer lung tissues (control) and lung cancer patients showed that, although protein levels of PD-L1 and phospho-GSK3α (i.e., inhibited GSK3α) were strongly elevated in tumor tissues, ARIH1 protein levels were higher in control samples (Supplementary Fig. 7). This finding is consistent with our discovery that ARIH1 promotes PD-L1 degradation and suggest that loss of ARIH1 expression in cancer is a mechanism of PD-L1 accumulation that drives escape from anti-tumor immunity. The mechanisms that result in decreased ARIH1 expression levels in cancer remain to be elucidated.

Notably, ARIH1 mutation found in large cell lung carcinoma (Y392C)[35] blocked ARIH1-induced PD-L1 ubiquitination and degradation (Fig. 6a, b), indicating that mutational inactivation of ARIH1 in cancer could lead to accumulation of PD-L1, to promote escape from anti-tumor immunity. This observation suggests a potential mechanism of why this mutation occurs in clinical patients with large cell lung carcinoma.

We established a 4T1 cell line with stable ARIH1-OE to evaluate the role of ARIH1 in PD-L1 degradation and tumorigenesis. In accord with our previous findings, ARIH1-OE reduced the protein levels of PD-L1 (Fig. 6c). ARIH1 over-expression had no effect on cell proliferation in vitro (Supplementary Fig. 8a) and on tumor growth in immunodeficient nude

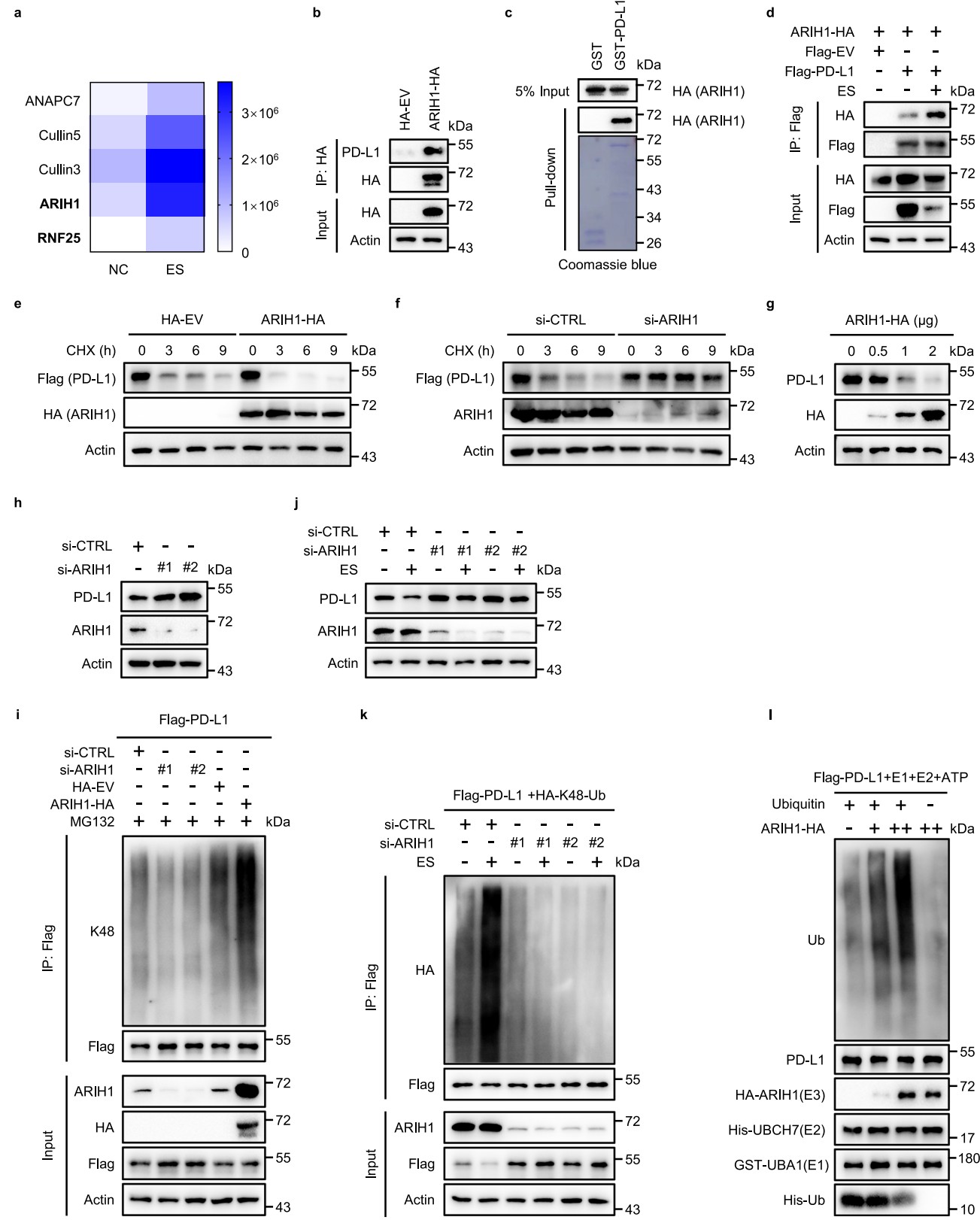

mice (Supplementary Fig. 8b–d). However, we observed a dramatically suppressed tumor growth in the ARIH1-OE group in immunocompetent BALB/c mice, the majority of which exhibited a complete tumor regression (Fig. 6d–f). The levels of total and activated CD8[+] cytotoxic T cells (GzmB[+]) that infiltrated the tumor microenvironment were significantly

increased in the ARIH1-OE group (Fig. 6g, h and Supplementary Fig. 9). Consistently, ARIH1 overexpression resulted in an increased expression of inflammatory cytokines, including IFNγ, tumor necrosis factor-α, and T-cell chemokines C-C motif chemokine ligand 5 (CCL-5) and C-X-C motif chemokine ligand 10 (CXCL-10), as judged by quantitative reverse transcription

**Fig. 4 ARIH1 mediates PD-L1 ubiquitination and degradation. a** As in Fig. 3b, except ubiquitination-related proteins are shown in the heatmap.
**b** Co-IP analysis for the interaction of ARIH1 and endogenous PD-L1 in HEK293T cells transfected with HA-ARIH1; HA-tagged empty vector (HA-EV) was transfected as a negative control. **c** Recombinant PD-L1 was purified using a GST pull-down assay and incubated with HEK293T lysates, which were transfected with HA-ARIH1. The interaction between ARIH1 and PD-L1 was detected by Immunoblot assay. **d** Co-IP analysis for the interaction of ARIH1 and PD-L1 in HEK293T cells transfected with HA-ARIH1 and Flag-PD-L1, treated with or without 10 µM ES-072 for 24 h; Flag-tagged empty vector (Flag-EV) was transfected as a negative control. **e, f** PD-L1 level in 20 µg/mL cycloheximide (CHX)-treated HEK293T cells transfected with or without HA-ARIH1 (**e**) and ARIH1-siRNA (**f**). **g, h** Immunoblots of PD-L1 and ARIH1 (HA) in H1975 cells transfected with ARIH1-HA (**g**) or ARIH1-siRNAs (**h**). **i** HEK293T cells were transfected with Flag-PD-L1. Co-IP analysis for the interaction of K48-ubiquitin and PD-L1 in HEK293T cells transfected with ARIH1-siRNAs or HA-ARIH1 and treated with MG132 (10 µM, a proteasome inhibitor) for 6 h. **j** Immunoblots of PD-L1 and ARIH1 in U937 cells transfected with ARIH1-siRNAs, following treatment with 10 µM ES-072 for 24 h. **k** HEK293T cells were transfected with HA-K48-ubiquitin and Flag-PD-L1. Co-IP analysis for the interaction of K48-ubiquitin and PD-L1 in HEK293T cells transfected with ARIH1-siRNAs and treated with or without 10 µM ES-072 for 24 h. **l** Recombinant PD-L1 and ARIH1 were purified in transfected HEK293T cells, respectively. An in vitro ubiquitination assay of PD-L1 was performed with purified GST-UBA1 (E1), His-UBCH7 (E2) in the presence or absence of ubiquitin or HA-ARIH1 (E3). Source data are provided as a Source Data file.

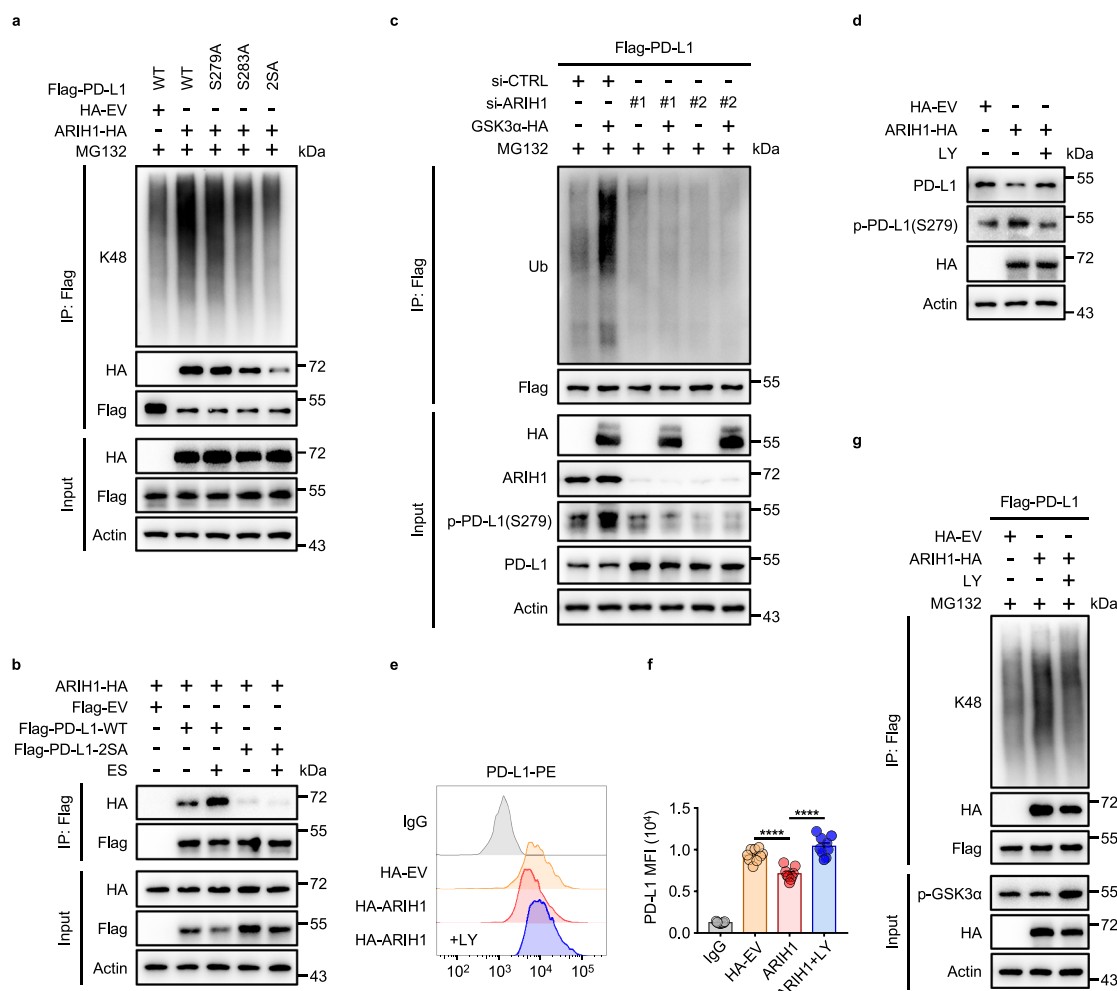

**Fig. 5 GSK3α-mediated phosphorylation of PD-L1 promotes PD-L1-ARIH1 interaction and ARIH1-induced degradation. a** Co-IP analysis for the interaction of K48-ubiquitin, ARIH1 (HA), and PD-L1 in HEK293T cells transfected with HA-ARIH1 and Flag-PD-L1 (WT, S279A, S283A, or 2SA), treated with MG132 (10 µM) for 6 h; HA-tagged empty vector (HA-EV) was transfected as a negative control. **b** Co-IP analysis for the interaction of ARIH1 and PD-L1 in HEK293T cells transfected with HA-ARIH1 and Flag-PD-L1 (WT or 2SA), treated with 10 µM ES-072 for 24 h, Flag-tagged empty vector (Flag-EV) was transfected as a negative control. **c** HEK293T cells were transfected with Flag-PD-L1. Co-IP analysis for the interaction of ubiquitin and PD-L1 in HEK293T cells transfected with ARIH1-siRNAs or HA-GSK3α and treated with MG132 (10 µM) for 6 h; non-targeting siRNA (si-CTRL) was transfected as a negative control. **d** Immunoblots of PD-L1 and ARIH1 (HA) in H1975 cells transfected with HA-ARIH1, treated with or without 5 µM LY for 6 h. **e, f** MFI (**e**) and relative quantification (**f**) of PD-L1 in HA-ARIH1-overexpressed H1975 cells, treated with or without 5 µM LY for 6 h. Data represent means ± SEM, n = 9, 3 independent repeats, ****P < 0.0001. **g** HEK293T cells were transfected with Flag-PD-L1. Co-IP analysis for the interaction of K48-ubiquitin, ARIH1 (HA), and PD-L1 in HEK293T cells transfected with HA-ARIH1, treated with or without 5 µM LY for 12 h and treated with MG132 (10 µM) for 6 h. Source data are provided as a Source Data file.

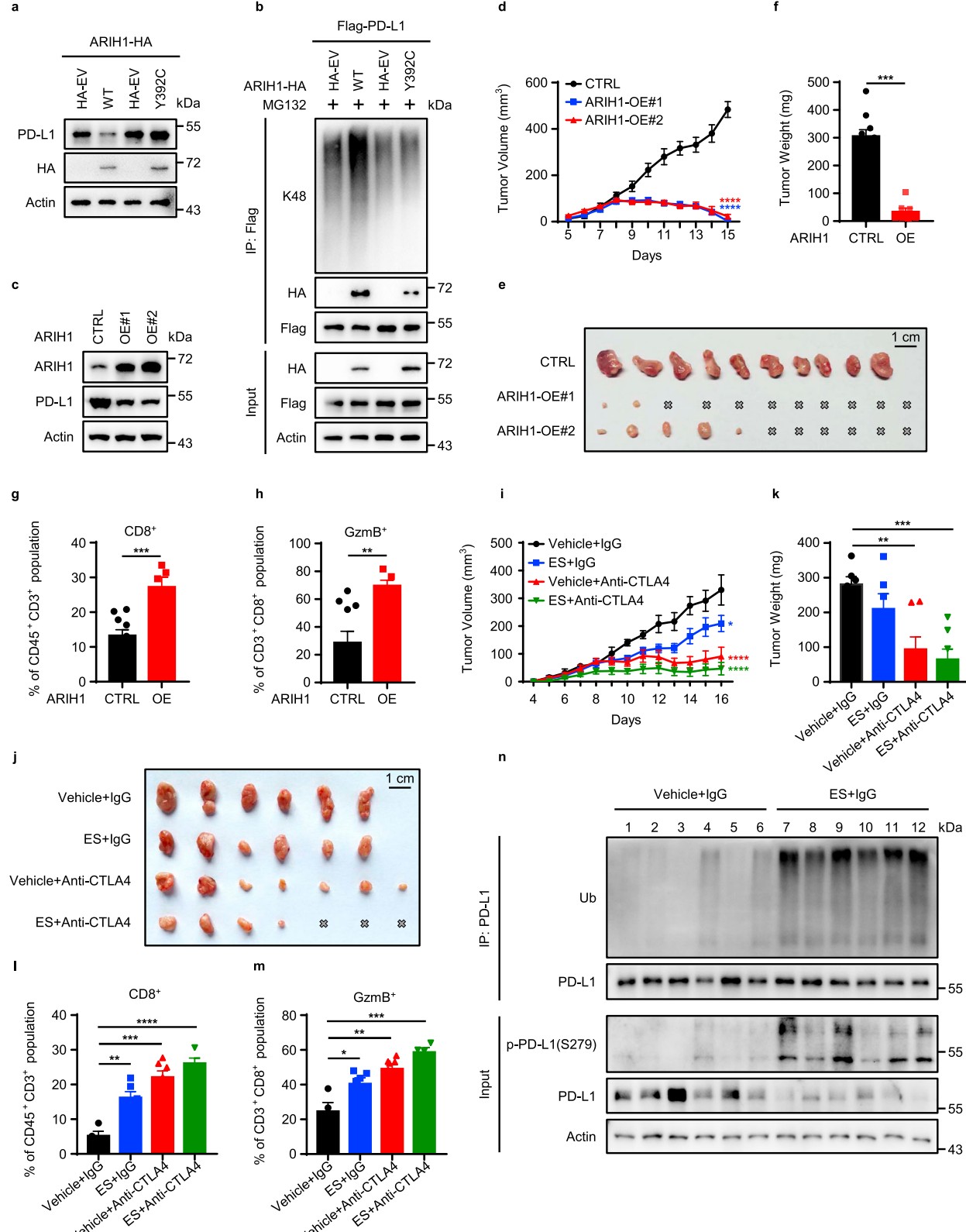

PCR (qRT-PCR) of the whole tumor mRNA (Supplementary Fig. 10a–d).

Recent studies have explored to intercept two immune checkpoint pathways, PD-L1 and CTLA4, collectively, to improve the efficacy of immunotherapy[36]. Next, we used the 4T1 tumor xenograft model and performed an anti-CTLA4 immune checkpoint blockade treatment in the presence/absence of

ES-072. Tumor growth was significantly decreased both by the ES-072 treatment and the anti-CTLA4 treatment. Notably, a further decrease of tumor growth and even complete regression was observed when ES-072 and anti-CTLA4 treatments were combined (Fig. 6i–k). The levels of total and activated CD8[+] cytotoxic T cells (GzmB[+]) in the tumors were also significantly increased upon the ES-072/anti-CTLA4 combination treatment

**Fig. 6 ARIH1 promotes anti-tumor immunity via PD-L1 degradation. a** Immunoblot of PD-L1 and ARIH1 (HA) in H1975 cells transfected with HA-ARIH1 (WT or Y392C); HA-tagged empty vector (HA-EV) was transfected as a negative control. **b** Co-IP analysis for the interaction of K48-ubiquitin, ARIH1 (HA), and PD-L1 in HEK293T cells transfected with Flag-PD-L1 and HA-ARIH1 (WT or Y392C) in the presence of 10 μM MG132 for 6 h. **c** 4T1 cells were infected with an empty vector (CTRL) or two different ARIH1-overexpressing lentiviral preparations (OE#1 and OE#2). ARIH1 and PD-L1 levels were determined by immunoblotting. **d–f** Tumor growth (**d**, **e**) of CTRL ($n = 10$) and ARIH1-OE cells ($n = 11$) in BALB/c mice and final tumor weights (**f**). Data represent means ± SEM, ***$P < 0.001$ ($P = 0.0001$), ****$P < 0.0001$. **g**, **h** Flow cytometry analysis for the tumor levels of CD8$^+$ T cells (**g**) and CD8$^+$GzmB$^+$ T cells (**h**). Data represent means ± SEM, CTRL ($n = 10$) and ARIH1-OE ($n = 5$), ***$P < 0.001$ ($P = 0.0006$), **$P < 0.01$ ($P = 0.0046$). **i–k** 4T1 tumor xenograft growth in BALB/c mice (**i**, **j**) and final tumor weights (**k**) following treatment with ES-072 and/or anti-CTLA4 ($n = 6$-7). Vehicle = sodium carboxymethyl (CMC-Na). ES = ES-072. Data represent means ± SEM, *$P < 0.05$ ($P = 0.01$), **$P < 0.01$ ($P = 0.0013$), ***$P < 0.001$ ($P = 0.0001$), ****$P < 0.0001$. **l**, **m** Flow cytometry analysis for the tumor levels of CD8$^+$ T cells (**l**) ($n = 4$-7) and CD8$^+$GzmB$^+$ T cells (**m**) ($n = 4$-7). Data represent means ± SEM. **l** **$P < 0.01$ ($P = 0.0017$), ***$P < 0.001$ ($P = 0.0001$), ****$P < 0.0001$. **m** *$P < 0.05$ ($P = 0.0286$), **$P < 0.01$ ($P = 0.0012$), ***$P < 0.001$ ($P = 0.00099$). **n** Immunoblotting of the indicated tumor lysates. Source data are provided as a Source Data file.

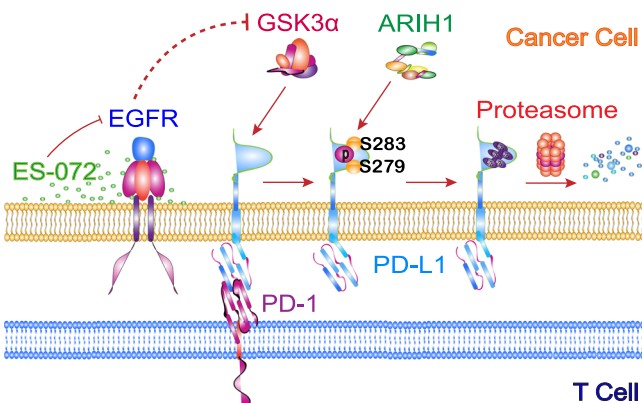

**Fig. 7 Schematic model for GSK3α-promoted and ARIH1-mediated PD-L1 degradation.** GSK3α phosphorylates PD-L1 at Ser279 and Ser283. This phosphorylation promotes the binding of PD-L1 with ARIH1, leading to PD-L1 ubiquitination and proteasomal degradation. This image was created by the first author.

when compared to either of the single treatments (Fig. 6l, m). Furthermore, western blotting analysis of the tumor lysates showed that ES-072 treatment decreased PD-L1 levels, while increasing its phosphorylation and ubiquitylation levels (Fig. 6n).

Taken together, our data indicate that ARIH1 plays a role in promoting anti-tumor immunity, and that ES-072 is a promising agent to boost anti-tumor immunity and anti-CTLA4 immune checkpoint blockade immunotherapies.

## Discussion

Our study identified that ES-072, an EGFR inhibitor, is an anti-PD-L1 agent that induces a signaling cascade downstream of GSK3α activation. Following ES-072 treatment, GSK3α phosphorylates PD-L1 at Ser279 and Ser283, which promotes recruitment of the E3 ubiquitin ligase ARIH1 that marks PD-L1 for proteasomal degradation (Fig. 7). Our tumor xenograft experiments revealed an important function of ARIH1 in promoting anti-tumor immunity. This finding indicates that various cancer therapies, including immune checkpoint blockade and cell-based immunotherapies, can be enhanced by supplementation of this inhibitor into the treatment regimens, as not only will it drive EGFR inhibition, decreasing cancer growth, but it will also drive degradation of PD-L1, enhancing anti-tumor immunity.

Recent studies show that PD-L1 expression and degradation levels are regulated by several protein tyrosine kinases, which seem to converge on the PI3K-AKT-GSK3 pathway, yet the mechanisms of PD-L1 degradation regulation were not fully clear[37–39]. Our findings suggest that the EGFR-AKT-GSK3α-ARIH1 axis is critical for the regulation of PD-L1 degradation.

These results also provide a broad insight into how cancer cells with RTK-activating mutations that drive inhibition of GSK3 may promote escape from anti-tumor immunity by preventing PD-L1 degradation and place RTK inhibitors that result in activation of GSK3 as putative enhancers of the PD-1/PD-L1 immune checkpoint blockade immunotherapies. Our work may pave the way for the augmentation of current immunotherapies that involve targeting PD-1/PD-L1 and help to overcome resistance to such therapies.

GSK3α and GSK3β share 97% amino acid similarity in their kinase domains[40]. The isoforms have both unique and overlapping functions, and one isoform cannot completely compensate for the loss function of another one. The most significant evidence for this is that GSK3α-deficient mice show increased insulin sensitivity[41], whereas GSK3β deletion in mice results in late embryonic death[42]. Growth factors, such as EGF, result in inactivation of GSK3 through Ser9 phosphorylation of GSK3α or Ser21 phosphorylation of GSK3β by AKT[43,44].

GSK3β has been reported to promote degradation of PD-L1 via phosphorylation at extracellular T180 and S184, contributing to anti-tumor immunity[8]. These two phosphorylation sites are located in the extracellular region of PD-L1, and their control by GSK3β is not fully understood. In this study, we found that GSK3α, but not GSK3β, co-immunoprecipitated with glycosylated PD-L1 following EGFR inhibition by ES-072 (Supplementary Fig. 4a–c). Our mechanistic studies showed that GSK3α phosphorylates PD-L1 at Ser279/283, at the cytosolic region of the receptor (Fig. 2c). The dual-mode of PD-L1 regulation by GSK3α via its cytosolic region and GSK3β via its extracellular region highlights the importance of oncogenic signaling pathways that converge on the EGFR-AKT-GSK3α/β axis in suppressing PD-L1 degradation and therefore promoting cancer escape from anti-tumor immunity[45–47]. Whether GSK3α or GSK3β on PD-L1 regulation are the only critical factors remains to be determined.

GSK3 is a tumor suppressor, with important roles in both solid tumors and blood tumors[48–51], where it regulates cancer cell viability and proliferation[52,53]. Aberrant expression of GSK3α in lung cancer has prognostic significance for clinical treatment[54]. In light of these reports and findings of our study, GSK3-activating agents are likely to promote a strong PD-L1 degradation phenotype and inhibit cancer escape from anti-tumor immunity, thereby enhancing various cancer therapies, including immunotherapies.

We identified ARIH1 as a promoter of anti-tumor immunity via induction of PD-L1 degradation. Previously, ARIH1 was shown to be elevated in head and neck squamous cell carcinoma biopsies[55]. However, we found that in lung alveolar adenocarcinoma biopsies, ARIH1 levels are lower than in control samples. This intriguing cancer-specific difference invites further investigation in future studies, to determine how it relates to the escape

from each cancer type from anti-tumor immunity. Interestingly, ARIH1 missense mutations are found in 4% of non-small cell lung cancer patients ($n = 75$)[56], 3.25% of prostate cancer patients ($n = 154$)[57], and 5.13% of cutaneous squamous cell carcinoma patients ($n = 39$)[58].

In addition to ARIH1, PD-L1 is also targeted for degradation by Cul3[SPOP] and β-TrCP[7,8] E3 ubiquitin ligases. Together, these findings suggest that combination treatments that target more than one of these E3 ubiquitin ligases may have potentially additive/synergistic effects on PD-L1 degradation in cancer, allowing a more potent stimulation of anti-tumor immunity.

In summary, our study sheds light on the mechanisms of cancer escape from anti-tumor immunity via increased PD-L1 protein levels downstream of EGFR overexpression and over-activation in cancer. Our work suggests that GSK3α- and ARIH1-activating agents, as well as EGFR inhibitors, are likely to stimulate anti-tumor immunity and therefore enhance existing cancer therapies by triggering the PD-L1 degradation pathway this work delineates.

## Methods

**Reagents and antibody generation**. The compounds and their sources are as follows: S-Ruxolitinib (#INCB018424; Selleck), AZD9291, Osimertinib (#S7297; Selleck), MG132 (#S2619; Selleck), LY2090314 (#S7063; Selleck), and ES-072 were synthesized in collaboration with Shanghai Institute of Organic Chemistry, Chinese Academy of Sciences. The recombinant cytokines and their sources are as follows: Human recombinant IFNγ (#300-02; Peprotech), Human recombinant EGF (#AF-100-15; Peprotech), and Mouse recombinant IL-4 (#214-14; Peprotech). The following antibodies were used in this study: PE anti-human CD274 (#329706; 1 : 200; Biolegend), PE anti-mouse CD274 (#124308; 1 : 200; Biolegend), PE Mouse IgG2b (isotype control) (#400312; 1 : 200; Biolegend), Zombie Violet™ Fixable Viability Kit (#423114; 1 : 200; Biolegend), PerCP/Cyanine5.5 anti-mouse CD45 (#103132; 1 : 200; Biolegend), PE/Cyanine7 anti-mouse CD3 (#100320; 1 : 200; Biolegend), FITC anti-mouse CD8 (#100706; 1 : 200; Biolegend), APC anti-human/mouse Granzyme B (#372204; 1 : 200; Biolegend), PD-L1 (ab213524, 1 : 1000; Abcam), PD-L1 (66248-1-Ig, 1 : 1000; Proteintech), β-TrCP (D13F10) (#4394, 1 : 1000; Cell Signaling Technology), EGFR (D38B1, 1 : 1000; Cell Signaling Technology), Phospho-EGFR (Tyr1068, 1 : 1000; Cell Signaling Technology), Ubiquitin (P4D1) (#SC-8017, 1 : 200; Santa Cruz Biotechnology), K48 (05-1307, 1 : 1000; Milipore), GSK3α (#4337, 1 : 1000; Cell Signaling Technology), Phospho-GSK3α (Ser21) (#9631, 1 : 1000; Cell Signaling Technology), GSK3β (Y174) (ab32391, 1 : 5000; Abcam), Phospho-GSK3β (Ser9) (#P49841, 1 : 1000; Cell Signaling Technology), AKT (#9272, 1 : 1000; Cell Signaling Technology), Phospho-AKT (Ser473) (#4046, 1 : 2000; Cell Signaling Technology), ARIH1 (C-7) (#SC-514551, 1 : 200; Santa Cruz Biotechnology), ARIH1 (Goat) (#EB05812, 1 : 100; Everestbiotech), GST (B-14) (#SC-138, 1 : 200; Santa Cruz Biotechnology), Granzyme B (D6E9W) (#46890, 1 : 50; Cell Signaling Technology), His-tag (#66005-1-Ig, 1 : 1000; Proteintech), Flag-tag (0912-1, 1 : 2000; HuaAn Biotechnology), HA-tag (0906-1, 1 : 2000; HuaAn Biotechnology), and β-Actin (M1210-2, 1 : 2000; HuaAn Biotechnology). The anti-human phospho-PD-L1 Ser279 antibody was raised against the region near Ser279 phosphorylation site of PD-L1. The secondary antibodies for western blotting were used: goat anti-mouse (1 : 20,000, #31430, Thermo Fisher Scientific, Ltd) and goat anti-rabbit (1 : 20,000, #31460, Thermo Fisher Scientific, Ltd). The phosphorylated synthetic peptide [QDTNSKKQSDTHLEC] was used for immunization in rabbits. The antibody was generated by GenScript (Nanjing, China). Anti-HA magnetic beads (#B26202) and Anti-Flag (DYKDDDDK) Affinity Gel (#B23102) were from Bimake. Lipofectamine 2000 (#1901433) and Lipofectamine 3000 (#2067450) were from Invitrogen. Collagenase/hyaluronidase (#17100-017, Vancouver, BC, Canada) were from Stemcell Technologies and DNase (#10104159001) were from Sigma. The peptides and their sources are as follows: PD-L1-WT [LRKGRMMDVKKCGIQDTNSKKQSDTH-LERT], phosphorylated PD-L1-WT [LRKGRMMDVKKCGIQDTNS (PO3H2) KKQS (PO3H2) DTHLERT], and PD-L1-2SA [LRKGRMMDVKKCGIQDTNAKKQADTH-LEET] were synthesized by Zhongtai (Hangzhou, China).

**Cell culture**. Human histiocytic lymphosarcoma cell line (U937), Non-small cell lung adenocarcinoma cancer cell line (H1975), and Human embryonic kidney cell line (HEK293T) were a gift from J.Y. Yuan (Harvard Medical School, Boston). PDMs were obtained from male BALB/c mice. PDMs, U937, and H1975 cells were cultured in RPMI-1640 (Hyclone, with L-glutamine); HEK293T cells were cultured in Dulbecco's modified Eagle's medium (Hyclone, with L-glutamine, with 4.5 g/L glucose, without pyruvate). These media were supplemented with 10% heat-inactivated fetal calf serum (Gibco) and 1% Penicillin/Streptomycin (Gibco). All cells were cultured at 37 °C with 5% CO$_2$. Cells were transiently transfected with DNA using Lipofectamine 2000 or Lipofectamine 3000.

**Plasmids construction and RNA interference**. Flag-PD-L1 (WT and mutants), GSK3α-HA, and ARIH1-HA (WT and mutants) were amplified by PCR and fused

the fragments into pCMV3 via seamless cloning. The constructs and siRNAs were transfected into cell lines (HEK293T, H1975, and U937) with Lipofectamine 2000 or Lipofectamine 3000, according to the the manufacturer's protocol. siRNAs used in this study are provided in Supplementary Table 5.

**In vitro high-throughput drug screens**. U937 cells ($2 \times 10^5$) were preincubated with 100 μg/L IFNγ for 48 h and plated in 96-well plates per well (Corning). Then 10 μM FDA-approved drugs or drug candidates were added and incubated for 12 h. All compounds were commercially purchased. Treatments were performed twice; each plate contained a negative control (dimethyl sulfoxide) and a positive control (Ruxolitinib). U937 cells were centrifuged for 5 min at $1000 \times g$ and the supernatant discarded. The collected cells were washed with phosphate-buffered saline (PBS) twice and incubated with PE anti-human CD274 at 4 °C for 30 min. Then the incubated U937 cells were washed and resuspended with PBS, and the protein level of membrane PD-L1 reflected by PD-L1-PE median fluorescence intensity (MFI) was determined using flow cytometry analysis. The hit compounds were picked and classified according to the PD-L1-PE-MFI and the targeted pathways.

**Western blot analysis**. For western blot analysis, cells were collected and washed with PBS, then lysed in radioimmunoprecipitation assay (RIPA) buffer (1% Triton X-100, 100 mM Tris-HCl pH 8.8, 100 mM NaCl, 0.5 mM EDTA). After incubation on ice for 30 min, the lysates were centrifuged at $12,000 \times g$ for 10 min at 4 °C. The supernatant was collected and the protein concentration was measured by bicinchoninic acid reaction. Protein samples were added with 2× loading buffer and heated at 100 °C for 10 min, separated with SDS-polyacrylamide gel electrophoresis (SDS-PAGE), transferred onto polyvinylidene difluoride membranes, and blocked with 5% non-fat milk in Phosphate Buffered Saline with 0.1% Tween-20 (PBST) for 1 h at room temperature. The membranes were probed with the corresponding primary antibodies at 4 °C overnight and horseradish peroxidase-conjugated secondary antibodies at room temperature for 1 h. Signals were detected using chemiluminescence reagents (#4AW001-500, 4A649 Biotech, Co.).

**Immunoprecipitation**. For immunoprecipitation between PD-L1 and GSK3α/ARIH1/K48, cells were lysed in RIPA buffer (1% Triton X-100, 100 mM Tris-HCl pH 8.8, 100 mM NaCl, 0.5 mM EDTA) supplemented with a complete protease inhibitor cocktail (Bimake, added fresh), and mixed with antibodies at 4 °C for 4 h; protein A/G agarose beads were added and incubated at 4 °C overnight. Beads were washed three times with RIPA buffer and subjected to western blotting.

**Mass spectrometry and data analysis**. Flag-tagged PD-L1 was overexpressed in HEK293T cells and were trypsin-digested. PD-L1 was immunoprecipitated with beads following immunoprecipitation. The resulting peptides were subjected to the phosphopeptide enrichment using TiO$_2$ beads. The enriched phospho-peptides were analyzed on the Q Exactive™ HF mass spectrometer (Thermo Fisher Scientific). The identification and quantification of phosphorylated peptides were done by MaxQuant. The tandem mass spectra were searched against the UniProt human protein database together with a set of commonly observed contaminants. The precursor mass tolerance was set as 20 p.p.m. and the fragment mass tolerance was set as 0.1 Da. The 33 cysteine carbamide methylation was set as a static modification and the methionine oxidation, as well as serine, threonine, and tyrosine phosphorylation, were set as variable modifications. The false discovery rate (FDR) at peptide spectrum match level was controlled below 1%.

**Duolink® PLA fluorescence analysis**. For the interaction between PD-L1 and GSK3α/GSK3β, the samples were pre-treated with respect to fixation, retrieval, and/or permeabilization. Then the samples were incubated with Duolink® Blocking Solution in a heated humidity chamber for 60 min at 37 °C. The samples were incubated with diluted primary antibody in a humidity chamber overnight at 4 °C. The samples were washed with 1× Wash Buffer A at room temperature for 5 min twice after the primary antibody solution was moved then incubated with diluted PLUS, and MINUS PLA probes (1 : 5) in a pre-heated humidity chamber for 1 h at 37 °C. The samples were washed with 1× Wash Buffer A at room temperature for 5 min twice after the PLA probes were moved, then incubated with ligation solution in a pre-heated humidity chamber for 30 min at 37 °C. The samples were washed with 1× Wash Buffer A at room temperature for 5 min twice after the ligation solution was moved then incubated with amplification solution in a pre-heated humidity chamber for 100 min at 37 °C. The samples were washed with 1× Wash Buffer B for 10 min twice and 0.01x Wash Buffer B for 1 min at room temperature after the amplification solution was moved. The slides were mounted with a coverslip using a minimal volume of Duolink® PLA Mounting Medium with 4′,6-diamidino-2-phenylindole analyzed with confocal microscope. The images were collected using Cytation 3 and were analyzed using Gen5 2.0.

**Protein purification and in vitro kinase assays**. For purification of PD-L1, Flag-tagged PD-L1 was transfected into HEK293T cells for 24 h. Cells were lysed in 1 mL of lysis buffer (TAP) (0.5% NP-40, 1 mM Na$_3$VO$_4$, 20 mM Tris-HCl pH 7.5, 1 mM NaF, 150 mM NaCl, 1 mM EDTA) supplemented with a complete protease inhibitor cocktail (Bimake, added fresh), and incubated with anti-Flag magnetic

beads (after washing the beads with PBS twice) for 6 h on a rotating wheel at 4 °C. The beads were washed with TAP buffer three times and treated with CIP (#M0290, Biolabs) at 37 °C for 30 min. The kinase assays were performed with recombinant human GSK3α proteins (#Ab42597, Abcam). The purified PD-L1 or synthetic peptides (PD-L1-WT/2SA) were incubated in 30 μL of kinase buffer (25 mM Tris-HCl pH 7.5, 5 mM β-glycerophosphate, 2 mM dithiothreitol (DTT), 0.1 mM $Na_3VO_4$, 10 mM $MgCl_2$) supplemented with phosphatase inhibitor cocktail (#b15001, Bimake), with or without 100 μM ATP for 2 h at 37 °C. The reactions were stopped by adding SDS-PAGE 2× loading buffer (100 mM Tris-HCl pH 6.8, 20% glycerol, 4% SDS, 0.1% Bromophenol blue 0.2 M DTT) and heating at 100 °C for 10 min. Kinase activity was evaluated by dot blot or western blotting with anti-phospho-human PD-L1 Ser279 antibody.

**Flow cytometry analysis of membrane PD-L1.** For flow cytometric analysis for membrane PD-L1, H1975 or HEK293T cells were collected by centrifugation at $1000 \times g$ for 5 min, incubated with PBS (0.5% bovine serum albumin) for 10 min at room temperature. The cells were probed with PE-conjugated PD-L1 antibody (#329706, Biolegend) and a matched isotype control at 4 °C for 30 min in the dark. After washing three times with PBS, the cells were analyzed using flow cytometry (Beckman Coulter Cytoflex) and data were analyzed using CytExpert V2.3 and and FlowJo X software.

**GST-pull down.** GST-tagged PD-L1 was expressed in *Escherichia coli* BL21 and purified. HA-tagged ARIH1 was expressed in HEK293T cells and purified using magnetic HA beads. For reaction, purified GST-PD-L1 was first incubated with glutathione-Sepharose 4B beads at 4 °C for 1 h. Then, the HA-tagged protein was added and incubated at 4 °C for 2 h. The mixture was boiling with 1× loading buffer for 10 min, then subjected to western blotting.

**In vitro ubiquitination assays.** Plasmids GST-UBA1 (E1), Flag-PD-L1, HA-ARIH1 (E3), and ARIH1 inactive mutant were transfected into HEK293T cells. Post transfection, cells were collected and lysed in RIPA Lysis Buffer at 4 °C for 1 h. Then, the lysates were incubated with indicated beads at 4 °C overnight. His-tagged protein UBCH7 (E2) ubiquitin was purified by *E. coli* BL21 expression. Reactions were performed in a 30 μL reaction mixture at 37 °C for 2 h in the presence of His-Ub, E1, E2, E3, Flag-PD-L1, ATP regeneration solution (Enzo Life Sciences) and Ubiquitin Reaction Buffer (Enzo Life Sciences). All reactions were terminated by boiling 10 min with SDS sample buffer and then subjected to western blotting.

**Immunohistochemistry.** EGFR-WT tumors and EGFR-mutant-driven tumors from primary lung adenocarcinoma tissues were obtained from 8 patients (4 cases each group, median age: 60 years old, range from 47 to 81) at the Sir Run Run Shaw Hospital, Zhejiang University. The four EGFR mutations are L858R point mutation of exon 21; E542K point mutation of exon 10 and deletion of exon 19 (ex19del 745–750); P753R point mutation of exon 19; L858R point mutation of exon 21. All samples were collected with signed informed consent according to the internal review and ethics boards of Sir Run Run Shaw Hospital. These tissues were rapidly excised, fixed with 4% paraformaldehyde, and embedded in paraffin for tissue sections (5 μm thick) and immunohistochemical staining. The primary antibodies used are anti-PD-L1 (1 : 200), anti-p-PD-L1 (Ser279) (1 : 200), anti-p-GSK3α (1 : 200), and anti-ARIH1 (1 : 200). Visualization of cell nuclei was performed with hematoxylin and analysis was done using the Olympus BX61 light microscope.

**Generation of ARIH1-OE stable 4T1 cell lines.** Briefly, HEK293T cells were transfected with PCDH-CTRL and PCDH-mouse ARIH1 with packaging plasmids. Medium with secreted viruses was collected at 48, 72, and 96 h, and was filtered through 0.45 μm filters. Twenty-four hours post infection, the medium was replaced with fresh medium and the infected 4T1 cells were selected with 4 μg/mL puromycin for 3 days.

**Tumor xenograft experiments.** Female BALB/c mice or nude mice (aged 8–10 weeks) were purchased from Shanghai SLAC Laboratory Animal, Co., Ltd (Shanghai, China). All the animal experiments were strictly conducted in accordance with the protocols approved by the Ethics Committee for Animal Studies at Zhejiang University, China.

For xenograft model with control or ARIH1-stable 4T1 cells, $5 \times 10^5$ control or ARIH1-stable 4T1 cells suspended in 50 μL PBS and Matrigel (1 : 1 v/v) were injected into the fourth breast fat pad. On days 3–5 after injection, tumor size was measured and calculated by using the formula $1/2 \times$ length $\times$ width$^2$. Tumor weight was recorded on the day of killing.

For xenograft model with 4T1 cells and combination therapy with ES-072 and anti-CTLA4, $5 \times 10^5$ 4T1 cells suspended in 50 μL PBS and Matrigel (1 : 1 v/v) were injected into the fourth breast fat pad. On days 3 after injection, tumor size was measured and calculated by using the formula $1/2 \times$ length $\times$ width$^2$. Mice were then randomly divided into control group: IgG (100 μg/100 μL, intraperitoneal injection, i.p.) and 0.5% sodium carboxymethyl (200 μL, intragastric administration, i.g) treatment; ES-072 treatment group: IgG (100 μg/100 μL, i.p.) and ES-072 (60 mg/kg, 200 μL, i.g.); anti-CTLA4 treatment group: anti-CTLA4 (100 μg/100 μL, i.p.) and 0.5%

sodium carboxymethyl (200 μL, i.g.); ES-072 and anti-CTLA4 combination group: anti-CTLA4 (100 μg/100 μL, i.p.) and ES-072 (60 mg/kg, 200 μL, i.g.). ES-072 was given daily from 3 days after inoculation and anti-CTLA4 antibody was administered on day 7, 10, and 13 after inoculation with respective control treatment.

**Tumor sample preparation and flow cytometry.** Excised tumors were digested in collagenase/hyaluronidase and DNase at 37 °C for 45 min to make cell suspension with a 45 μm filter (BD Bioscience). Then, cells were stained with Percp-Cy5.5-conjugated-CD45, PE-Cy7-conjugated-CD8, FITC-conjugated-CD3 antibodies, fixed and permeabilized with a Fix/Perm kit (Biolegend), and finally stained with APC-conjugated-GzmB antibody. Data acquisition was performed using flow cytometry (Beckman Coulter Cytoflex) and data were analyzed using CytExpert V2.3 software.

**Immunoprecipitation assay with mouse tumor tissues.** For immunoprecipitation between PD-L1 and ubiquitin, fresh xenograft tissues were lysed in RIPA buffer (1% Triton X-100, 100 mM Tris-HCl pH 8.8, 100 mM NaCl, 0.5 mM EDTA) supplemented with a complete protease inhibitor cocktail (Bimake, added fresh) and mixed with anti-PD-L1 at 4 °C for 4 h; protein A/G agarose beads were added and incubated at 4 °C overnight. Beads were washed three times with RIPA buffer and subjected to western blotting.

**qRT-PCR analysis for tumor cytokines.** Fresh tumor tissues were lysed and total RNA was extracted using TRIzol™ Plus RNA Purification Kit (Invitrogen). cDNA was synthesized from purified RNA using the PrimeScript RT reagent Kit (TAKARA, RR047A) according to the manufacturer's instructions. Quantitative PCR was performed using a StepOnePlus Real-Time PCR Systems (ABI). The comparative Ct method was used for the data analysis and mouse β-actin mRNA was used as an internal control. The sequences of primers used for qRT-PCR are provided in Supplementary Table 6.

**Statistics and reproducibility.** Numerical data are presented as means ± SEM; all data analyses were performed using GraphPad Prism (version 7.0, GraphPad Software, Inc.). Unpaired Student's $t$-test was used to analyze the flow cytometry data. One-way analysis of variance was used to analyze the statistical differences among the groups with $P$-values indicated in the related graphs. The level of statistical significance was set at a $p < 0.05$. All assays were carried out at least three independent times with the same results.

**Reporting summary.** Further information on research design is available in the Nature Research Reporting Summary linked to this article.

## Data availability

The data that support the findings of this study are available from the corresponding author upon reasonable request. The mass spectrometry proteomics data have been deposited to the ProteomeXchange Consortium (http://proteomecentral.proteomexchange.org) via the iProX partner repository with the dataset identifier PXD024452. Source data are provided with this paper.

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

## Acknowledgements

This work was supported by the National Natural Science Foundation of China (number 81773182, 91854108, 31601121, 81870007, 81920108001, and 81800024), the National Key R&D Program of China (2017YFA0104200), Zhejiang Provincial Program for the Cultivation of High-Level Innovative Health Talents (2016-63) and Zhejiang Provincial Natural Science Foundation (LD19H160001). We thank Professor Hui Yang from the Institute of Neuroscience, Chinese Academy of Sciences for cDNA, Professor Qiming Sun from Zhejiang University for his helpful suggestions to this project.

## Author contributions

H.X. conceived and coordinated the project. H.X., S.Y., and A.N. designed the experiments, interpreted the data, and wrote the manuscript. Y.W., C.Z., X.L., and Z.H. performed most of the experiments and interpreted the data. B.S., Q. Zeng, Q. Zhao, H.Z., H.L., X.C., X.X., M.Z., T.H., Z.W., H.Y., S. Yang, Y.S., Y.C., R.W., T.X., and W.C. assisted with the experiments and helped to analyze the data. H.X., A.N., S. Ying, Y.W., and C.Z. revised the manuscript.

## Competing interests

The authors declare no competing interests.
