## [Peer Review File · Nature Communications]

REVIEWER COMMENTS

Reviewer #1 (Remarks to the Author):

In this study, the investigators sought to identify new mechanisms of PD-L1 degradation by performing a high throughput screen using >2000 FDA-approved drugs. Using this screening approach, they identified several EGFR inhibitors as inducers of PD-L1 degradation. Upon treatment with EGFR inhibitors, they detail a mechanism whereby EGFR inactivation activates GSK3 α , which promotes phosphorylation of PD-L1 at the intracellular S279 and S283 residues, and that this phosphorylation event is necessary for the interaction between PD-L1 and ARIH1, thereby leading to proteasome-mediated degradation of PD-L1. Additionally, they present human data demonstrating that tumors with EGFR mutations have lower levels of ARIH1, compromised PD-L1 ubiquitination, and decreased immunogenicity. This study is novel because the investigators identify ARIH1 as a novel E3 ligase responsible for ubiquitylation of PD-L1. These findings also suggest that EGFR inhibitors that result in GSK3 α activation may improve the efficacy of PD-1/PD-L1 immune checkpoint blockade. However, several issues need to be addressed to merit publication in Nature Communications. For example, the dosage being used for their EGFR inhibitor ES-072 makes it difficult to determine if the results are due specifically to EGFR inhibition or off-target effects. The paper sites that the drug has an IC50 of 1.75 nM for the T790M/L858R mutant, however it was used at roughly ~1000 times that concentration before the desired effect on PD-L1 was observed. Additional in vitro kinase assays to determine if the drug affects GSK3 activity directly would alleviate some of the concerns behind their proposed mechanism. Major and minor comments are included below:

Major Comments:

1. Fig. 2d-f are confusing. Why are the investigators using FLAG-tagged PD-L1 in experiments shown in Fig. 2d-e when there are good antibodies that recognize endogenous PD-L1?
2. In Fig. 2f, the investigators say they want to confirm that endogenous PD-L1 is phosphorylated following ES-072 treatment. To do this, they generated a phospho-specific antibody against PD-L1 Ser279. Why do total PD-L1 levels not change in the western shown in Fig. 2f? This is concerning given the results of Fig. 2D and Fig. 1e and h. Additionally, they are using extremely high doses of ES-072 in these experiments and should validate findings with lower doses.
3. While the IP Flag westerns shown in Fig. 2g are supportive, how was the phospho-specific antibody validated for immunohistochemistry shown in Fig. 2h? A western blot for phospho- and total PD-L1 levels should be performed in tumors with WT and mutant EGFR to see if the trend persists. Also, this should be repeated with a larger sample size (1 tumor is not sufficient).
4. Fig. 3: To highlight the difference in the GSK3 isoforms, the investigators should knockdown or knockout only the beta isoform. This would strengthen the claim that it is only alpha that phosphorylates PD-L1.
5. Figure 3d: Are there any changes to the GSK3 beta isoform activity? This would act as an additional control.
6. Figure 6 a, d-e IHC data: How many tumors were analyzed? Are these images representative of a larger sample size? Quantification such as H-score analysis should be reported for a number of tumors.
7. The majority of experiments were performed with ES-072 (the investigators say this is due to the stability and specificity of the drug). However, it is important to know whether other FDA approved EGFR inhibitors such as Osimertinib also activate GSK3 α .

Minor Comments:

1. Line 124-126 in the manuscript states that "knockdown of EGFR in H1975 cells blocked the decrease of cell membrane PD-L1 levels induced by ES-072", and then references Supplementary Fig. 3h. This is not what is shown in the figure.
2. Heat maps shown in Fig 1a, 3a, and 4a need a better description in the figure legend. Information provided is not sufficient. Similarly, the map of the MS/MS analysis shown in Fig. 2a should be better described.
3. Figure 5c. The investigators should add a blot showing PD-L1 phosphorylation to demonstrate that this is in fact due to differences in the ARIH1 levels only, and that there aren't off-target

effects of ARIH1 KD on PD-L1 phosphorylation.
4. The overall grammar can be improved.

Reviewer #2 (Remarks to the Author):

This is an interesting study, and the authors show a new mechanism of PD-L1 degradation via EGFR/GSK3/ARIH1 signaling. The authors also suggest ARIH1 as a novel E3 ligase for PD-L1, which could lead to the development of therapeutic strategy in cancers with EGFR mutations. However, some points should be addressed as the comments below.

Although most in vitro experiments well conducted and the result acquired revealed a significant post-translational regulation axis of PD-L1, the overall novelty of this manuscript may not reach Nat Comm's scope.

1. Based on the hypothesis and drug-screen design, the authors are interested in how membrane PD-L1 is degraded (internalized). However, the evidence they provided afterward only looked at the whole protein regardless of subcellular localization. If the authors believe that "dynamic ubiquitin modification of PD-L1 for proteasomal degradation should mostly occur in the intracellular segment of PD-L1", they may need to perform microscopy or membrane protein labeling to look at the PD-L1 level change and PD-L1-GSK3a interaction on cell membrane specifically.

2. In the previous report (PMID: 27572267), non-glycosylated PD-L1 could interact with GSK3b, and then GSK3b causes PD-L1 phosphorylation. On the contrary, the proposed mechanism here is glycosylation-independent theoretically, making it possible to answer how membrane-localized (glycosylated) PD-L1 is degraded. Considering the similarity of GSK3a and GSK3b, it would be better to see whether PD-L1/GSK3a binding is also glycosylation-dependent or not.

3. The in vivo experiments showed here are relatively superficial to support that EGFR inhibition promotes antitumor immunity by degrading PD-L1 (compared to published articles of similar topics). We should expect to see 1) Tumor development of PD-L1 WT vs. 2SA model on both immune-competent and -compromised mice. 2) Combination of EGFR inhibitor and non-PD-1/PD-L1 blockade immunotherapy (i.e., IL-2, CTLA-4 mAb, etc.).

Reviewer #3 (Remarks to the Author):

In this manuscript, the authors proposed a new regulatory mechanism of PD-L1 degradation in cancer cells. Using a high-throughput drug screening system, the authors found several tyrosine kinase inhibitors could reduce PD-L1 levels. Inhibition of EGFR promoted PD-L1 phosphorylation at intracellular Ser279 and Ser283 sites by GSK3a, leading to ARIH1-mediated PD-L1 ubiquitination and proteasomal degradation. I have the following concerns about data interpretation and several suggestions to improve data quality and clarity.

1. The central finding of this manuscript is ES-072 can induce PD-L1 degradation. Western blot data in Figure 1c showed that ES-072 treatment caused almost full degradation of PD-L1, at least to a level much lower than that under mock treatment (no IFNg, no ES-072). However, in Figure 1b, ES-072 treatment only caused moderate decrease of surface PD-L1 level. More importantly, PD-L1 level under IFNg+ES-072 treatment in Figure 1b was much higher than that under mock treatment without IFNg+ES-072. How can the authors explain this discrepancy?

2. The authors concluded that Ser279/283 phosphorylation plays a central role in ES-072-induced PD-L1 degradation. However, in Figure 2d and I, PD-L1 2SA mutant still showed substantial degradation under ES-072 treatment.

3. The authors stated that GSK3a knockdown rescued PD-L1 degradation induced by ES-072, which was not true according to Figure 3h. In general, GSK3a-knockdown cells had higher PD-L1 levels but ES-072 induced PD-L1 degradation was still substantial.

4. The authors claimed that ARIH1 promoted anti-tumor immunity via PD-L1 degradation, but the current data are insufficient to support their conclusion. A tumor experiment with manipulation of ARIH1 level is needed to confirm the role of ARIH1 in anti-tumor immunity.
5. The authors found tumor infiltration of GzmB+ T cells was increased after oral administration of ES-072 in a 4T1-luciferase syngeneic tumor xenograft model. More biochemical and immunological analysis of the tumor microenvironment is needed. Levels of PD-L1 phosphorylation, ubiquitination and degradation need to be checked. Tumor-infiltrated immune cells need to be carefully assessed.
6. It shall be interesting to test if ES-072 can synergize with anti-PD-1 or anti-PD-L1 in treating cancer.
7. As ARIH1 mutation has been found in large cell lung carcinoma, is there any relationship between ARIH1 mutation and disease progression?
8. There is a large quantity of gel data in this paper but none of them has molecular weight markers.

Reviewers' comments:

Reviewer #1 (Remarks to the Author)

In this study, the investigators sought to identify new mechanisms of PD-L1 degradation by performing a high throughput screen using >2000 FDA-approved drugs. Using this screening approach, they identified several EGFR inhibitors as inducers of PD-L1 degradation. Upon treatment with EGFR inhibitors, they detail a mechanism whereby EGFR inactivation activates GSK3 α , which promotes phosphorylation of PD-L1 at the intracellular S279 and S283 residues, and that this phosphorylation event is necessary for the interaction between PD-L1 and ARIH1, thereby leading to proteasome-mediated degradation of PD-L1. Additionally, they present human data demonstrating that tumors with EGFR mutations have lower levels of ARIH1, compromised PD-L1 ubiquitination, and decreased immunogenicity. This study is novel because the investigators identify ARIH1 as a novel E3 ligase responsible for ubiquitylation of PD-L1. These findings also suggest that EGFR inhibitors that result in GSK3 α activation may improve the efficacy of PD-1/PD-L1 immune checkpoint blockade. However, several issues need to be addressed to merit publication in Nature Communications. For example, the dosage being used for their EGFR inhibitor ES-072 makes it difficult to determine if the results are due specifically to EGFR inhibition or off-target effects. The paper sites that the drug has an IC50 of 1.75 nM for the T790M/L858R mutant, however it was used at roughly ~1000 times that concentration before the desired effect on PD-L1 was observed. Additional *in vitro* kinase assays to determine if the drug affects GSK3 activity directly would alleviate some of the concerns behind their proposed mechanism. Major and minor comments are included below:

Response:

We thank the reviewer for their appreciation of our study and the constructive comments. We have now addressed the comments in detail below. We have also performed new experiments and added a total of 36 new experimental panels to the manuscript. The text, newly added to the manuscript, is indicated in red. We hope that our responses are adequate and we are open to further suggestions where our responses fall short. Please note that the figure citations in our responses below refer to the new (post-revision) figures.

The paper sites that the drug has an IC50 of 1.75 nM for the T790M/L858R mutant, however it was used at roughly ~1000 times that concentration before the desired effect on PD-L1 was observed. Additional *in vitro* kinase assays to determine if the drug affects GSK3 activity directly would alleviate some of the concerns behind their proposed mechanism.

Response:

We thank the reviewer for this suggestion. We have now performed additional *in vitro* kinase assays for GSK3 α activity with ES-072 and LY2090314 (GSK3 α/β inhibitor) and included the results in Supplementary Table 1 and below. The results of three independent tests showed IC50 of positive control (LY2090314) for GSK3 α are 0.98 nM, 0.81 nM and 0.84 nM respectively. And IC50 of ES-072 for GSK3 α is >40 μ M. Since the concentration of ES-072 used

in our experiments is 10 μ M, it would not affect GSK3 activity directly, supporting our proposed mechanism of ES-072-induced PD-L1 degradation being via EGFR-GSK3 α -ARH1 signaling.

ES-072 did not affect the activity of GSK3 α directly.

GSK3 α		
Compound ID	Operator	IC50 (nM)
ES-072	>	40000
LY2090314	=	0.98
ES-072	>	40000
LY2090314	=	0.81
ES-072	>	40000
LY2090314	=	0.84
Staurosporine	=	46
Staurosporine	=	46

Supplementary Table 1. *In vitro* kinase specificity profiling of ES-072 and LY2090314 for GSK3 α .

Major Comments:

1. Fig. 2d-f are confusing. Why are the investigators using FLAG-tagged PD-L1 in experiments shown in Fig. 2d-e when there are good antibodies that recognize endogenous PD-L1?

Response:

We thank the reviewer for this comment. In Fig. 2d and 2e, wild-type and the mutant versions (S279A, S283A and 2SA) of PD-L1 are examined to better demonstrate the function of two phosphorylation sites (S279 and S283) in regulating the stability of PD-L1. Therefore, for accurate comparison of wild-type versus mutant PD-L1, exogenous plasmids and Flag antibody were used. The PD-L1 antibody used in Fig 2f is an endogenous PD-L1 antibody.

Phosphorylation on S279/S283 impairs the stability of PD-L1.

Figure 2d. Immunoblots of PD-L1 (anti-Flag) in HEK293T cells transfected with Flag-tagged-PD-L1 (WT) and mutated Flag-PD-L1 on S279A, S283A or 2SA following treatment with 25

ng/mL EGF and/or 10 μ M ES-072 for 48 h, 2SA represents S279A/S283A.

Figure 2e. Immunoblots of PD-L1 (anti-Flag) in HEK293T cells transfected with Flag-tagged-PD-L1 (WT/2SA) following treatment with 20 μ g/mL cycloheximide (CHX) for indicated times.

Figure 2f. Immunoblots of H1975 cell lysates following ES-072 treatment for 2 h, at indicated doses.

2. In Fig. 2f, the investigators say they want to confirm that endogenous PD-L1 is phosphorylated following ES-072 treatment. To do this, they generated a phospho-specific antibody against PD-L1 Ser279. Why do total PD-L1 levels not change in the western shown in Fig. 2f? This is concerning given the results of Fig. 2D and Fig. 1e and h. Additionally, they are using extremely high doses of ES-072 in these experiments and should validate findings with lower doses.

Response:

We thank the reviewer for this comment. In Fig. 2f of original version, a shorter time point (2 h) was used to be able to compare total PD-L1 levels and analyze changes in its phosphorylation, as longer time points with lower concentrations of ES-072 result in PD-L1 degradation. However, to detect a robust change in PD-L1 phosphorylation at 2 h, we employed a higher concentration of ES-072. We have now repeated this experiment at 0.1 μ M and 0.5 μ M doses of ES-072 and included the results in Fig 2f and below.

ES-072 enhanced the phosphorylation of PD-L1 on S279.

Figure 2f. Immunoblots of H1975 cell lysates following ES-072 treatment for 2 h, at indicated doses.

3. While the IP Flag westerns shown in Fig. 2g are supportive, how was the phospho-specific antibody validated for immunohistochemistry shown in Fig. 2h? A western blot for phospho- and total PD-L1 levels should be performed in tumors with WT and mutant EGFR to see if the trend persists. Also, this should be repeated with a larger sample size (1 tumor is not sufficient).

Response:

We thank the reviewer for this comment. We also think this is an important control. However, as these are human lung adenocarcinoma specimens, we cannot employ a Ser/Ala control slide to validate the antibody for human IHC samples. Moreover, validating the antibody for a tumor xenograft sample will not be indicative of its performance in human tumor biopsy samples, which are more complex on the cellular level due to presence of an intact immunity. However, our Ser/Ala experiments using western blotting indicate that the antibody is specific. To support the IHC findings shown in Fig 2h, we have now analyzed the tumor lysates from human lung adenocarcinoma specimens (4 EGFR-WT samples and 4 EGFR-mutant samples) by western blotting using the p-PD-L1 and PD-L1 antibodies and included the results in Fig 2k and below. We also increased IHC sample size in Fig 2h and performed area density analysis for these samples in Fig 2i-j and below. Both western blotting of tumor cell lysates and immunohistochemistry analysis of tumor biopsies showed that PD-L1 phosphorylation levels at Ser279 are higher in EGFR-WT tumors compared to EGFR-mutant-driven tumors.

PD-L1 phosphorylation levels at Ser279 are higher in EGFR-WT tumors than that in EGFR-mutant-driven tumors.

Figure 2k. Immunoblots of p-PD-L1 (Ser279) and PD-L1 in EGFR wild-type (n=4) versus mutant (n=4) human lung adenocarcinoma specimens.

Figure 2h. Representative images of p-PD-L1 (Ser279) and PD-L1 immunohistochemistry (IHC) staining from EGFR wild-type versus mutant human alveolar adenocarcinoma specimens. Scale bars represent 50 μ m.

Figure 2i-j. Quantification of IHC analysis for p-PD-L1 (Ser279) (i) and PD-L1 (j) in h. The graph shows mean \pm S.E.M.; ** $p < 0.01$.

4. Fig. 3: To highlight the difference in the GSK3 isoforms, the investigators should knockdown or knockout only the beta isoform. This would strengthen the claim that it is only alpha that phosphorylates PD-L1.

Response:

We thank the reviewer for this comment. Both GSK3 isoforms (GSK3 α and GSK3 β) can phosphorylate PD-L1, but on different sites and depending on the glycosylation extent of PD-L1.

GSK3 α phosphorylates PD-L1 at S279 and S283 that are located in the intracellular segment of PD-L1 (this study), while GSK3 β phosphorylates PD-L1 at T180 and S184 that are located in the extracellular segment of PD-L1⁸. The existing data (Fig 3f-h, shown below) and newly added knockout experiments (Supplementary Fig. 4f, shown below) strongly indicate that GSK3 α , but not GSK3 β , phosphorylates PD-L1 at Ser279.

In addition, we used tunicamycin (TM), a N-linked glycosylation inhibitor, to remove glycosylation of PD-L1 in HEK293 cells and found that GSK3 α interacts with both glycosylated PD-L1 and non-glycosylated PD-L1, while GSK3 β only interacts with non-glycosylated PD-L1

(Supplementary Fig. 4a-c, shown below). We hope our experiments and explanations are sufficient to make the difference between the GSK3 isoforms clear.

8. Li, C. W. et al. Glycosylation and stabilization of programmed death ligand-1 suppresses T-cell activity. *Nat. Commun.* **7**, 12632 (2016).

GSK3 α , but not GSK3 β , phosphorylates PD-L1 at Ser279.

Figure 3f. The phosphorylation of PD-L1-WT peptides and mutated PD-L1-2SA (S279A/S283A) peptides by GSK3 α in an *in vitro* kinase assay, in the presence or absence of ATP. p-PD-L1 peptides were synthesized as a positive control. The phosphorylation of PD-L1 peptides was detected by dot blot with anti-p-PD-L1 (Ser279) antibody.

Figure 3g. *In vitro* GSK3 α kinase assay, was performed in HEK293T cells transfected with Flag-tagged PD-L1 and GSK3 α -HA. Total cell lysates were immunoprecipitated with anti-Flag or anti-HA. The phosphorylation of PD-L1 by GSK3 α was detected using an anti-p-PD-L1 (Ser279) antibody.

Figure 3h. Immunoblots of p-PD-L1, PD-L1, and GSK3 α in H1975 cells transfected with GSK3 α -siRNAs.

Supplementary Fig. 4f. Immunoblots of p-PD-L1, PD-L1, and GSK3 β in H1975 cells transfected with GSK3 β -siRNAs.

GSK3 α , but not GSK3 β , interacts with glycosylated PD-L1.

Supplementary Fig. 4a. Co-immunoprecipitation (Co-IP) analysis for the interaction of GSK3 α /GSK3 β and PD-L1 in HEK293T cells transfected with Flag-tagged-PD-L1 and treated with or without 5 μ M tunicamycin (TM, an N-linked glycosylation inhibitor) for 12 h, Flag-tagged empty vector (Flag-EV) was transfected as a negative control.

Supplementary Fig. 4b-c. Proximity ligation assay (PLA) analysis for the interaction of GSK3 α (b) or GSK3 β (c) and PD-L1 in HEK293T cells treated as a. PLA signals are shown in red and the nuclei in blue. Quantification for the mean area (MA) of PD-L1/GSK3 α or PD-L1/GSK3 β PLA speckles are indicated by scattergram. The graph shows mean \pm S.E.M.; NS: no significant; ****P < 0.0001 versus respective controls.

5. Figure 3d: Are there any changes to the GSK3 beta isoform activity? This would act as an additional control.

Response:

We thank the reviewer for this comment. Both isoforms of GSK3 are known to be activated following EGFR inhibition. Growth factors, such as EGF, result in inactivation of GSK3 through Ser9 phosphorylation of GSK3 α or Ser21 phosphorylation of GSK3 β by AKT^{43,44}. We have added the blots showing GSK3 β and p-GSK3 β in Fig 3e shown below.

43. Matsuda, T. et al. Distinct roles of GSK-3alpha and GSK-3beta phosphorylation in the heart under pressure overload. *Proc Natl Acad Sci U S A.* **105**, 20900-20905 (2008).

44. Ahmad, F. et al. Cardiomyocyte-specific deletion of Gsk3alpha mitigates post-myocardial infarction remodeling, contractile dysfunction, and heart failure. *J. Am. Coll. Cardiol.* **64**, 696-706 (2014).

Both isoforms of GSK3 are known to be activated following EGFR inhibition.

Figure 3e. Immunoblots of p-GSK3 α , GSK3 α , p-GSK3 β , GSK3 β , p-AKT, AKT, p-EGFR, and EGFR in H1975 cells treated with 10 μ M ES-072 for indicated times.

6. Figure 6 a, d-e IHC data: How many tumors were analyzed? Are these images representative of a larger sample size? Quantification such as H-score analysis should be reported for a number of tumors.

Response:

We thank the reviewer for this comment. We have now quantified the signals from the tumor IHC experiments with area density analysis and included them in Supplemental Figure 7 a-d and below. 3-4 tumors/normal tissues and 3-6 images from each sample were used to perform the quantification.

In addition, we merged the data of PD-L1 and p-PD-L1 (Ser279) in EGFR-WT tumors and EGFR-mutant-driven tumors into Fig 2h-j (see the response to comment #3).

Tumor samples exhibited higher PD-L1/p-GSK3 α and lower ARIH1 levels than normal samples.

Supplemental Figure 7a. Representative images of PD-L1, p-GSK3 α and ARIH1 IHC staining from human alveolar adenocarcinoma and paracancerous normal tissue specimens. Scale bars represent 20 μ m.

Supplemental Figure 7b-d. Quantification of IHC analysis for PD-L1 (b), p-GSK3 α (c), and ARIH1 (d). The graph shows mean \pm S.E.M.; *p < 0.05, **p < 0.01.

7. The majority of experiments were performed with ES-072 (the investigators say this is due to the stability and specificity of the drug). However, it is important to know whether other FDA approved EGFR inhibitors such as Osimertinib also activate GSK3 alpha.

Response:

We thank the reviewer for this comment. We chose three EGFR inhibitors, Gefitinib, AZD9291, and ES-072 and compared their effect on EGFR, GSK3 α and GSK3 β in Supplemental Figure 4e and below. ES-072 showed a higher activation effect on GSK3 α when compared to the other two EGFR inhibitors, as judged by loss of the inhibitory phosphorylation of GSK3 α .

Minor Comments:

1. Line 124-126 in the manuscript states that “knockdown of EGFR in H1975 cells blocked the decrease of cell membrane PD-L1 levels induced by ES-072”, and then references Supplementary Fig. 3h. This is not what is shown in the figure.

Response:

We thank the reviewer for this comment. We have now amended the text accordingly as below.

Furthermore, the decrease of PD-L1 levels induced by ES-072 is dependent on EGFR (Supplementary Fig. 3h).

2. Heat maps shown in Fig 1a, 3a, and 4a need a better description in the figure legend. Information provided is not sufficient. Similarly, the map of the MS/MS analysis shown in Fig. 2a should be better described.

Response:

We thank the reviewer for this comment. We have now amended the figure legends and expanded the descriptions of the indicated figure elements in the text and below.

Figure 1a. High-throughput screening of 2125 FDA-approved drugs. U937 cells were incubated with IFN γ (100 ng/mL) for 48 h, treated with the drugs at 10 μ M for 12 h. Ruxolitinib (Rux) was used as a positive control. The hit compounds that induced the decrease of PD-L1 levels are shown in blue. The heat map represents the targeted pathways obtained from the high throughput screening, based upon decreased membrane PD-L1 level detected by flow cytometry.

Figure 3b. HEK293T cells were transfected with Flag-PD-L1 and treated with ES-072 (10 μ M) for 48 h. Proteins that co-immunoprecipitated with Flag-PD-L1 were analyzed by mass spectrometry. Kinases and kinase-related proteins are shown in the heatmap.

Figure 4a. As in Figure 3b, except, ubiquitination-related proteins are shown in the heatmap.

Figure 2a. Mapping PD-L1 phosphorylation sites following ES-072 treatment. HEK293T cells were transfected with Flag-PD-L1 and treated with 10 μ M ES-072 for 48 h. Immunoprecipitated PD-L1 was analyzed by mass spectrometry following phosphopeptide enrichment. Peptide ionization data corresponding to Ser279/283 is shown.

3. Figure 5c. The investigators should add a blot showing PD-L1 phosphorylation to demonstrate that this is in fact due to differences in the ARIH1 levels only, and that there aren't off-target effects of ARIH1 KD on PD-L1 phosphorylation.

Response:

We thank the reviewer for this comment. This is an important control. We have now added the p-PD-L1(S279) blot to the results in Fig 5c and below.

Figure 5c. HEK293T cells were transfected with Flag-tagged PD-L1. Co-IP analysis for the interaction of ubiquitin and PD-L1 in HEK293T cells transfected with ARIH1-siRNAs or HA-tagged GSK3 α and treated with MG132 (10 μ M) for 6 h, non-targeting siRNA (si-CTRL) was transfected as a negative control.

4. The overall grammar can be improved.

Response:

We thank the reviewer for this comment. We have now amended the text. The text, newly added and amended to the manuscript, is indicated in red.

Reviewer #2 (Remarks to the Author):

This is an interesting study, and the authors show a new mechanism of PD-L1 degradation via EGFR/GSK3/ARIH1 signaling. The authors also suggest ARIH1 as a novel E3 ligase for PD-L1, which could lead to the development of therapeutic strategy in cancers with EGFR mutations. However, some points should be addressed as the comments below.

Although most *in vitro* experiments well conducted and the result acquired revealed a significant post-translational regulation axis of PD-L1, the overall novelty of this manuscript may not reach Nat Comm's scope.

Response:

We thank the reviewer for their appreciation of our study and the constructive comments. We have now addressed the comments in detail below. We have also performed new experiments and added a total of 36 new experimental panels to the manuscript. The text, newly added to the manuscript, is indicated in red. We hope that our responses are adequate and are open to further suggestions where our responses fall short. Please note that the figure citations in our responses below refer to the new (post-revision) figures.

Our study delineates a mechanism of PD-L1 degradation via EGFR-GSK3 α -ARIH1 signaling and identifies ARIH1 as a new E3 ubiquitin ligase for PD-L1. ARIH1 may become a potential therapeutic target to boost anti-tumor immunity and enhance immune checkpoint blockade immunotherapies, as activating ARIH1 would decrease PD-L1 levels and thus, cancer escape from immunity.

As per reviewer's suggestions, we have now added two *in vivo* experiments further supporting our model.

1) Overexpression of ARIH1 dramatically suppressed tumor development in immunocompetent BALB/c mice, accompanied by a complete recession for a majority of tumors, as well as increased infiltration by CD8⁺ and GzmB⁺ T cells and increased expression of inflammatory cytokines and chemokines. ARIH1 overexpression did not affect cell proliferation *in vitro* and in immunodeficient nude mice. These data further suggest the role of ARIH1 in anti-tumor immunity (see response to comment #3).

2) We used an anti-CTLA4 monoclonal antibody in combination with ES-072 to treat a syngeneic mouse breast tumor xenograft of 4T1 cells. The combination group showed an additive effect in the drop of tumor size and increase of CD8⁺ and GzmB⁺ T cells. Our data indicates that ES-072 has the potential to promote anti-tumor immunity and improve the efficacy of anti-CTLA4 therapy *in vivo* (see response to comments #3).

1. Based on the hypothesis and drug-screen design, the authors are interested in how membrane PD-L1 is degraded (internalized). However, the evidence they provided afterward only looked at the whole protein regardless of subcellular localization. If the authors believe that "dynamic ubiquitin modification of PD-L1 for proteasomal degradation should mostly occur in the intracellular segment of PD-L1", they may need to perform microscopy or membrane protein labeling to look at the PD-L1 level change and PD-L1-GSK3a interaction on cell membrane specifically.

Response:

We thank the reviewer for this comment. Our screen is designed to detect both effects (internalization and degradation), however, as it is hard to predict the fate of an internalized receptor in this case, we are interested only in drug hits that upon follow up of the screen showed PD-L1 degradation, which is a more clinically useful effect than internalization.

In addition, we extracted the outer membrane protein (OMP) in H1975 cells to look at the PD-L1 level changes and PD-L1-GSK3a interaction on cell membrane. PD-L1 in the whole cell lysate (WCL) was detected from 34 kDa to 55 kDa because of glycosylation, but the majority of PD-L1 in outer membrane protein (OMP) was only detected at 55 kDa. PD-L1 level was decreased by ES-072 both in WCL and OMP. Then we immunoprecipitated PD-L1 from OMP and found a basal level of GSK3 α /PD-L1 interaction, which was enhanced by ES-072. We hope that this data addresses the issue.

ES-072 enhanced GSK3 α /PD-L1 interaction and decreased PD-L1 levels on cell membrane.

Figure a. H1975 cells were treated with or without 10 μ M ES-072 for 12 h. Outer membrane proteins (OMP) was extracted with biotin from whole cell lysate (WCL), PD-L1 was IP with anti-PD-L1 in outer membrane proteins (OMP) and immunoblot with GSK3 α and PD-L1.

2. In the previous report (PMID: 27572267), non-glycosylated PD-L1 could interact with GSK3b, and then GSK3b causes PD-L1 phosphorylation. On the contrary, the proposed mechanism here is glycosylation-independent theoretically, making it possible to answer how membrane-localized (glycosylated) PD-L1 is degraded. Considering the similarity of GSK3a and GSK3b, it would be better to see whether PD-L1/GSK3a binding is also glycosylation-dependent or not.

Response:

We thank the reviewer for this comment. We have now performed this experiment and added the data to Supplementary Fig. 4a-c and below.

In this experiment, we used tunicamycin (TM), a N-linked glycosylation inhibitor, to remove glycosylation of PD-L1 in HEK293 cells and found that GSK3 α interacts with both glycosylated PD-L1 and non-glycosylated PD-L1, while GSK3 β only interacts with non-glycosylated PD-L1 (Supplementary Fig. 4a-c, shown below). We hope our experiments and explanations are sufficient to make the difference between the GSK3 isoforms clear.

GSK3 α , but not GSK3 β , interacts with glycosylated PD-L1.

Supplementary Fig. 4a. Co-immunoprecipitation (Co-IP) analysis for the interaction of GSK3 α /GSK3 β and PD-L1 in HEK293T cells transfected with Flag-tagged-PD-L1 and treated with or without 5 μ M tunicamycin (TM, an N-linked glycosylation inhibitor) for 12 h, Flag-tagged empty vector (Flag-EV) was transfected as a negative control.

3. The *in vivo* experiments showed here are relatively superficial to support that EGFR inhibition promotes antitumor immunity by degrading PD-L1 (compared to published articles of similar topics). We should expect to see 1) Tumor development of PD-L1 WT vs. 2SA model on both immune-competent and -compromised mice. 2) Combination of EGFR inhibitor and non-PD-1/PD-L1 blockade immunotherapy (i.e., IL-2, CTLA-4 mAb, etc.).

Response:

We thank the reviewer for this comment. The most significant role of two phosphorylation sites (S279 and S283) of PD-L1 is to regulate the interaction between PD-L1 and ARIH1. Therefore, we think it is more important to emphasize the role of ARIH1 in tumor models on both immune-competent and immune-compromised mice. In addition, the combination of EGFR inhibitor (ES-072) and non-PD-1/PD-L1 blockade immunotherapy (CTLA-4 mAb) was performed in mouse breast syngeneic tumor xenografts of 4T1 cells. We have now performed a series of experiments and added the data to Fig 6d-m and Supplementary Fig 8-9.

1) Tumor development of 4T1-ARIH1-CTRL vs. 4T1-ARIH1-OE in immune-compromised and immune-competent mice.

ARIH1 overexpression (OE) had no effect on cell proliferation *in vitro* (Supplementary Fig. 8a), and in immunodeficient nude mice (Supplementary Fig. 8b-d). However, we observed a dramatically suppressed tumor growth in ARIH1-OE group in immunocompetent BALB/c mice, the majority of which exhibited a complete recession (CR) (Fig. 6d-f). The levels of total and activated CD8⁺ cytotoxic T cells (GzmB⁺) that infiltrated the tumor microenvironment were significantly increased in the ARIH1-OE group (Fig. 6g-h). Consistently, compared with control tumors, tumors from the ARIH1-OE group displayed increased expression of inflammatory cytokines including IFN γ , TNF α , and T cell chemokines CCL-5 and CCL-10 (Supplementary Fig.9a-d).

ARIH1 overexpression showed no effect on proliferations of 4T1 cells *in vitro* and in immune-compromised mice.

Supplementary Fig. 8a. 4T1 cells were infected with an empty vector (CTRL) or two different ARIH1 overexpressing retroviral preparations (OE#1 and OE#2). Cell viability was monitored at indicated time points by an ATP assay. The graph shows the mean \pm SEM; NS: not significant.

Supplementary Fig. 8b-d. Tumor growth (b-c) of CTRL (n=4) and ARIH1-OE cells (n=7) in nude mice and final tumor weights (d). The graph shows the mean \pm SEM; NS: not significant.

ARIH1 overexpression suppressed 4T1 cell proliferations and increased immune response in immune-competent mice.

Fig. 6d-f. Tumor growth (d-e) of CTRL (n=10) and ARIH1-OE cells (n=11) in BALB/c mice and final tumor weights (f). The graph shows mean \pm S.E.M.; *** $P < 0.001$, **** $P < 0.0001$.

Figure 6g-h. Flow cytometry analysis for the tumor levels of CD8⁺ T cells (g) and CD8⁺GzmB⁺ T cells (h). The graph shows mean \pm S.E.M.; ** $p < 0.01$, *** $P < 0.001$.

Supplementary Fig. 9a-d. qRT-PCR analysis for the gene expression of IFN- γ (a), TNF- α (b), CCL-5 (c), and CCL-10 (d). The graph shows mean \pm SEM; * $p < 0.05$.

2) Combination of EGFR inhibitor (ES-072) and non-PD-1/PD-L1 blockade immunotherapy (CTLA-4 mAb).

We performed an anti-CTLA4 treatment in combination with ES-072 in a xenograft model of 4T1 cells. Tumor growth was significantly decreased both by oral administration of ES-072 and anti-CTLA4 treatments. Notably, a further decrease of tumor growth and even complete recession (CR) was observed in ES-072 and anti-CTLA4 combination group (Fig. 6i-k). Consistently, the levels of total and activated CD8⁺ cytotoxic T cells (GzmB⁺) in the tumors were also significantly increased upon the ES-072/anti-CTLA4 combination treatment, when compared to either of the single treatments (Fig. 6l-m).

ES-072 promoted anti-tumor immunity and improved the efficacy of anti-CTLA4 therapy *in vivo*

Fig. 6i-k. 4T1 tumor xenograft growth in BALB/c mice (i-j) and final tumor weights (k) following treatment with ES-072 and/or anti-CTLA4 (n=6-7). vehicle = sodium carboxymethyl (CMC-Na). ES = ES-072. The graph shows mean \pm S.E.M.; *p < 0.05, **p < 0.01, ***P < 0.001, ****P < 0.0001.

Fig. 6l-m. Flow cytometry analysis for the tumor levels of CD8⁺ T cells (l) and CD8⁺GzmB⁺ T cells (m). The graph shows mean \pm S.E.M.; *p < 0.05, **p < 0.01, ***P < 0.001, ****P < 0.0001.

Reviewer #3 (Remarks to the Author):

In this manuscript, the authors proposed a new regulatory mechanism of PD-L1 degradation in cancer cells. Using a high-throughput drug screening system, the authors found several tyrosine kinase inhibitors could reduce PD-L1 levels. Inhibition of EGFR promoted PD-L1 phosphorylation at intracellular Ser279 and Ser283 sites by GSK3a, leading to ARIH1-mediated PD-L1 ubiquitination and proteasomal degradation. I have the following concerns about data interpretation and several suggestions to improve data quality and clarity.

Response:

We thank the reviewer for their appreciation of our study and the constructive comments. We have now addressed the comments in detail below. We have also performed new experiments and added a total of 36 new experimental panels to the manuscript. The text, newly added to the manuscript, is indicated in red. We hope that our responses are adequate and are open to further suggestions where our responses fall short. Please note that the figure citations in our responses below refer to the new (post-revision) figures.

1. The central finding of this manuscript is ES-072 can induce PD-L1 degradation. Western blot data in Figure 1c showed that ES-072 treatment caused almost full degradation of PD-L1, at least to a level much lower than that under mock treatment (no IFN γ , no ES-072). However, in Figure 1b, ES-072 treatment only caused moderate decrease of surface PD-L1 level. More importantly, PD-L1 level under IFN γ +ES-072 treatment in Figure 1b was much higher than that under mock treatment without IFN γ +ES-072. How can the authors explain this discrepancy?

Response:

We thank the reviewer for this comment. We have repeated these experiments and the results still showed the same trend, we think the possible reasons for this phenomenon are:

1) Different detection methods and antibodies - flow cytometry (FCM) is a quantitative detection method and the PE-conjugated antibody (Clone 29E.2A3) is reported to recognize an epitope on PD-L1 within the PD-L1-CD80 binding region (PMID: 23918985). It should be noted that Clone 29E.2A3 does not work for western blotting (PMID: 26546452). While the FCM-based detection relies on the native form of membranal PD-L1, the antibody used in western blotting (WB) recognizes the primary structure of protein after heat denaturation and detects both membranal and cytosolic forms of the protein. The sensitivity of the two methods is also different.

2) Different target protein forms - PD-L1 exists as different forms due to its glycosylation modification. PD-L1 on the extracellular membrane is mostly highly glycosylated⁸. When analyzed by FCM, only the PD-L1 on the cell surface can be detected (single component, most of them are glycosylated PD-L1), When analyzed by WB, the whole protein of cell lysates

(complex components including non-glycosylated PD-L1, glycosylated PD-L1 and nuclear PD-L1) can be detected.

3) Different regulatory mechanisms - Interferon-gamma (IFN γ) induces PD-L1 expression via IFNGR-JAK-STAT pathway^{23,24}, EGFR inhibitors (ES-072 and AZD9291) induce PD-L1 proteasomal degradation mainly through the regulation on phosphorylation and ubiquitination levels. The differences may exist in the regulation of membrane PD-L1 and intracellular whole PD-L1 between these two mechanisms.

Although affected by several methodological factors, the central finding that ES-072 induces PD-L1 degradation is highly reproducible.

8. Li, C. W. et al. Glycosylation and stabilization of programmed death ligand-1 suppresses T-cell activity. *Nat. Commun.* **7**, 12632 (2016).
23. Horiuchi, M. et al. Interferon-gamma induces AT(2) receptor expression in fibroblasts by Jak/STAT pathway and interferon regulatory factor-1. *86*, 233-240 (2000).
24. Lau, T. S., Chan, K. Y., Cheung, T. H., Yim, S. F. & Kwong, J. Abstract 648: Interferon-gamma induces PD-L1 expression via IFNGR-JAK-STAT pathway in ovarian cancer. *Cancer Res.* **77**, 648 (2017).

2. The authors concluded that Ser279/283 phosphorylation plays a central role in ES-072-induced PD-L1 degradation. However, in Figure 2d and I, PD-L1 2SA mutant still showed substantial degradation under ES-072 treatment.

Response:

We thank the reviewer for this comment. There are a few reasons for this phenomenon:

1) In addition to the sites we found, there are other phosphorylation sites (T180 and A184 have been published⁸ and the other sites we found shown in Supplementary Table 3) that may also be important for degradation of PD-L1. Therefore, S2783/283A mutation is not sufficient to fully block PD-L1 degradation induced by ES-072. Future studies will reveal the kinases responsible for phosphorylation of those additional regulatory phospho-sites.

2) In order to describe our observations more accurately, we have now changed our wording in the manuscript shown below. We hope that these responses clarify the issue.

In agreement with the data which shows that Ser279/Ser283 phosphorylation is important for ES-072-induced PD-L1 degradation.

8. Li, C. W. et al. Glycosylation and stabilization of programmed death ligand-1 suppresses T-cell activity. *Nat. Commun.* **7**, 12632 (2016).

3. The authors stated that GSK3a knockdown rescued PD-L1 degradation induced by ES-072, which was not true according to Figure 3h. In general, GSK3a-knockdown cells had higher PD-L1 levels but ES-072 induced PD-L1 degradation was still substantial.

Response:

We thank the reviewer for this comment. There are a few reasons for this phenomenon:

1) In our opinion, this is because GSK3a knockdown is not as potent as a knockout would be and even residual GSK3a levels that remain after the knockdown may be activated enough by EGFR inhibition following ES-072 treatment to provide some phosphorylation of PD-L1 at Ser279. However, the effect of ES-072 on PD-L1 levels seen in lane 2 is clearly rescued by GSK3a knockdown (lanes 4, 6, and 8).

2) EGFR not only regulates protein degradation, but also regulates PD-L1 synthesis at the transcriptional level (JAK / STAT pathway), although we can make a certain rescue in the degree of degradation for PD-L1 by knocking down GSK3 α , the transcription factors still play their respective regulatory roles.

3) In addition to the kinase we found, there are other kinases (such as GSK3 β) that perform roles in PD-L1 degradation downstream EGFR inhibition. Knockdown of GSK3 α only is not enough to totally rescue the degradation of PD-L1 induced by ES-072.

4) In order to describe our observations more accurately, we have now changed our wording in the manuscript shown below. We hope that these responses clarify the issue.

Knockdown of GSK3 α in U937 and H1975 cells partially rescued PD-L1 degradation induced by ES-072 (Fig. 3i and Supplementary Fig. 4g).

8. Li, C. W. et al. Glycosylation and stabilization of programmed death ligand-1 suppresses T-cell activity. *Nat. Commun.* **7**, 12632 (2016).

4. The authors claimed that ARIH1 promoted anti-tumor immunity via PD-L1 degradation, but the current data are insufficient to support their conclusion. A tumor experiment with manipulation of ARIH1 level is needed to confirm the role of ARIH1 in anti-tumor immunity.

Response:

We thank the reviewer for the comment. We have now performed ARIH1 overexpression in tumor models on both immune-competent and immune-compromised mice and added the data to Fig 6d-h, Supplementary Fig. 8-9 and below.

ARIH1 overexpression (OE) had no effect on cell proliferation *in vitro* (Supplementary Fig. 8a), and in immunodeficient nude mice (Supplementary Fig. 8b-d). However, we observed a dramatically suppressed tumor growth in ARIH1-OE group in immunocompetent BALB/c mice, the majority of which exhibited a complete recession (CR) (Fig. 6d-f). The levels of total and

activated CD8⁺ cytotoxic T cells (GzmB⁺) that infiltrated the tumor microenvironment were significantly increased in the ARIH1-OE group (Fig. 6g-h). Consistently, compared with control tumors, tumors from the ARIH1-OE group displayed increased expression of inflammatory cytokines including IFN γ , TNF α , and T cell chemokines CCL-5 and CCL-10 (Supplementary Fig.9a-d).

ARIH1 overexpression showed no effect on proliferations of 4T1 cells *in vitro* and in immune-compromised mice.

Supplementary Fig. 8a. 4T1 cells were infected with an empty vector (CTRL) or two different ARIH1 overexpressing retroviral preparations (OE#1 and OE#2). Cell viability was monitored at indicated time points by an ATP assay. The graph shows the mean \pm SEM; NS: not significant.

Supplementary Fig. 8b-d. Tumor growth (b-c) of CTRL (n=4) and ARIH1-OE cells (n=7) in nude mice and final tumor weights (d). The graph shows the mean \pm SEM; NS: not significant.

ARIH1 overexpression suppressed 4T1 cell proliferations and increased immune response in immune-competent mice.

Fig. 6d-f. Tumor growth (d-e) of CTRL (n=10) and ARIH1-OE cells (n=11) in BALB/c mice and final tumor weights (f). The graph shows mean \pm S.E.M.; *** $P < 0.001$, **** $P < 0.0001$.

Figure 6g-h. Flow cytometry analysis for the tumor levels of CD8+ T cells (g) and CD8+GzmB+ T cells (h). The graph shows mean \pm S.E.M.; ** $p < 0.01$, *** $P < 0.001$.

Supplementary Fig. 9a-d. qRT-PCR analysis for the gene expression of IFN- γ (a), TNF- α (b), CCL-5 (c), and CCL-10 (d). The graph shows mean \pm SEM; * $p < 0.05$.

5. The authors found tumor infiltration of GzmB+ T cells was increased after oral administration of ES-072 in a 4T1-luciferase syngeneic tumor xenograft model. More biochemical and immunological analysis of the tumor microenvironment is needed. Levels of PD-L1 phosphorylation, ubiquitination and degradation need to be checked. Tumor-infiltrated immune cells need to be carefully assessed.

Response:

We thank the reviewer for the comment. We have now checked more indicators of immunological analysis in new added animal tumor models, such as total and activated CD8⁺ cytotoxic T cells (GzmB⁺) infiltration, inflammatory cytokines and T cell chemokines in tumor microenvironment (see response to comment #4 and comment #6). In addition, the levels of PD-L1 phosphorylation, ubiquitination and degradation were also analyzed and added in FIG 6n (see response to comment #6).

6. It shall be interesting to test if ES-072 can synergize with anti-PD-1 or anti-PD-L1 in treating cancer.

Response:

We thank the reviewer for the comment. As ES-072 already significantly decreases tumor PD-L1 levels we did not see any additive effect between such anti-PD-L1 treatment and ES-072 (data not shown), thus, we performed an anti-CTLA4 immune checkpoint blockade treatment in combination with ES-072. We performed an anti-CTLA4 treatment in combination with ES-072 in a xenograft model of 4T1 cells. Tumor growth was significantly decreased both by oral administration of ES-072 and anti-CTLA4 treatments. Notably, a further decrease of tumor growth and even complete recession (CR) was observed in ES-072 and anti-CTLA4 combination group (Fig. 6i-k). Consistently, the levels of total and activated CD8+ cytotoxic T cells (GzmB+) in the tumors were also significantly increased upon the ES-072/anti-CTLA4 combination treatment, when compared to either of the single treatments (Fig. 6l-m). Furthermore, western blotting analysis of the tumor lysates showed that ES-072 treatment decreased PD-L1, while increasing phosphorylation and ubiquitylation levels (Fig. 6n).

ES-072 promoted anti-tumor immunity and improved the efficacy of anti-CTLA4 therapy *in vivo*

Fig. 6i-k. 4T1 tumor xenograft growth in BALB/c mice (i-j) and final tumor weights (k) following treatment with ES-072 and/or anti-CTLA4 (n=6-7). vehicle = sodium carboxymethyl (CMC-Na). ES = ES-072. The graph shows mean ± S.E.M.; *p < 0.05, **p < 0.01, ***P < 0.001, ****P < 0.0001.

Fig. 6l-m. Flow cytometry analysis for the tumor levels of CD8+ T cells (l) and CD8+GzmB+ T cells (m). The graph shows mean ± S.E.M.; *p < 0.05, **p < 0.01, ***P < 0.001, ****P < 0.0001.

7. As ARIH1 mutation has been found in large cell lung carcinoma, is there any relationship between ARIH1 mutation and disease progression?

Response:

We thank the reviewer for the comment. We could not find any statistical significance in patient survival with the ARIH1 missense mutation the reviewer mentioned (versus those with wild-type ARIH1), as the data was available only from 4 patients, this analysis was deemed not very informative (source: cBioportal.com).

8. There is a large quantity of gel data in this paper but none of them has molecular weight markers.

Response:

We thank the reviewer for the comment. We have now added molecular weight markers in the figures.

REVIEWERS' COMMENTS

Reviewer #1 (Remarks to the Author):

The authors have done a commendable job addressing reviewer comments and added a significant amount of new data. The revised manuscript merits publication in Nature Communications. I have one remaining minor issue:

1) The authors should report the table of 160 FDA approved drugs that reduce PD-L1 membrane staining. This could be added as a supplementary table.

Reviewer #2 (Remarks to the Author):

The authors significantly improved the quality and clarity by adding several experimental data in the revised manuscript. Also, all the questions that I had in the previous version were addressed well.

Reviewer #3 (Remarks to the Author):

In the revised manuscript, the authors provided additional evidence to address my previous concerns. The new animal data further support the functional importance of ARIH1 in antitumor immunity. These efforts are fully appreciated by this reviewer. However, concern #1-3 have not been directly addressed, as shown below.

#1. The authors' answer to this concern is not satisfying. As pointed out by the authors in the rebuttal, western blotting can indicate total protein level. It is not straightforward to understand nearly complete loss of total PD-L1 (shown by WB) only led to partial loss of surface PD-L1 (shown by FACS). In Fig 1b, total PD-L1 level under IFN γ +ES-072 condition was much lower than that under mock condition. In Figure 1c-d, the result was opposite.

#2. The authors explained the other phosphorylation sites in addition to Ser279/283 might cause PD-L1 degradation under ES-072 treatment. Experimental evidence needs to be included to support this idea.

#3. The authors proposed that insufficient depletion of GSK3 α by the knockdown strategy and contribution of other EGFR-regulated kinases could explain that ES-072 induced PD-L1 degradation in GSK3 α -knockdown cells was still substantial. Again, these possibilities need to be tested by experiments. If GSK3 α turns out to be only one of the players in PD-L1 degradation, their major conclusion will need to be revised - "Together, our results delineate a mechanism of PD-L1 degradation and cancer escape from immunity via EGFR-GSK3 α -ARIH1 signaling and identify GSK3 α and ARIH1 as potential therapeutic targets to boost anti-tumor immunity and enhance immune checkpoint blockade immunotherapies."

Reviewer #1 (Remarks to the Author):

The authors have done a commendable job addressing reviewer comments and added a significant amount of new data. The revised manuscript merits publication in Nature Communications.

Response:

We thank the reviewer for their appreciation of our study and the constructive comments.

I have one remaining minor issue:

1) The authors should report the table of 160 FDA approved drugs that reduce PD-L1 membrane staining. This could be added as a supplementary table.

Response:

Thank you for the suggestion. We have added the information of 160 FDA approved drugs or drug candidates that reduce membrane PD-L1 level in supplementary table 1.

Reviewer #2 (Remarks to the Author):

The authors significantly improved the quality and clarity by adding several experimental data in the revised manuscript. Also, all the questions that I had in the previous version were addressed well.

Response:

We thank the reviewer for their appreciation of our study and the constructive comments.

Reviewer #3 (Remarks to the Author):

In the revised manuscript, the authors provided additional evidence to address my previous concerns. The new animal data further support the functional importance of ARIH1 in antitumor immunity. These efforts are fully appreciated by this reviewer.

Response:

We thank the reviewer for their appreciation of our study and the constructive comments.

However, concern #1-3 have not been directly addressed, as shown below.

#1. The authorso answer to this concern is not satisfying. As pointed out by the authors in the rebuttal, western blotting can indicate total protein level. It is not straightforward to understand nearly complete loss of total PD-L1 (shown by WB) only led to partial loss of surface PD-L1 (shown by FACS). In Fig 1b, total PD-L1 level under IFN γ +ES-072 condition was much lower than that under mock condition. In Figure 1c-d, the result was opposite.

Response:

We thank the reviewer for this comment.

1) Our new data shown below indicates that most of the PD-L1 is cytosolic (Figure A). As most of the cellular PD-L1 is not located at the plasma membrane, but on endomembrane components in the cytosol, the dramatic decrease of total PD-L1 levels seen by western blotting following ES-072 treatment reflects the dramatic changes of the cytosolic PD-L1, while FACS can only detect the plasma membrane PD-L1 levels, which change less dramatically than the total levels.

This is in accord with reports that plasma membrane PD-L1 is more stable than cytosolic PD-L1, at least partially, due to the glycosylation⁸. Therefore, cytosolic PD-L1 is more readily targeted for degradation following treatment with ES-072 than plasma membrane PD-L1.

These factors may account for the observations that ES-072 leads to a dramatic decrease in total PD-L1 (which consist of mostly cytosolic PD-L1), but a less dramatic change of plasma membrane PD-L1 (which is more stable).

2) It is not reasonable to directly compare PD-L1 levels under IFN γ +ES-072 condition and mock condition in Fig 1b and Fig 1c-d. Firstly, Fig 1b shows the cytosolic plus plasma membrane levels of PD-L1 but Fig 1c-d only shows plasma membrane levels of PD-L1. In addition, there are two different treatment conditions: IFN γ and ES-072. IFN γ induces PD-L1 expression via IFNGR-JAK-STAT pathway^{23,24}, while ES-072 induces PD-L1 proteasomal degradation. These two treatments may result in not directly comparable effects on cytosolic plus plasma membrane (Fig 1b) versus plasma membrane only PD-L1 levels (Fig 1c-d). We hope that these responses clarify the issue.

8. Li, C. W. et al. Glycosylation and stabilization of programmed death ligand-1 suppresses T-cell activity. *Nat. Commun.* **7**, 12632 (2016).

23. Horiuchi, M. et al. Interferon-gamma induces AT(2) receptor expression in fibroblasts by Jak/STAT pathway and interferon regulatory factor-1. *J Biol Chem* **275**, 233-240 (2000).

24. Lau, T. S., Chan, K. Y., Cheung, T. H., Yim, S. F. & Kwong, J. Abstract 648: Interferon-gamma induces PDL1 expression via IFNGR-JAK-STAT pathway in ovarian cancer. *Cancer Res.* 77, 648 (2017).

#2. The authors explained the other phosphorylation sites in addition to Ser279/283 might cause PD-L1 degradation under ES-072 treatment. Experimental evidence needs to be included to support this idea.

Response:

We thank the reviewer for this comment.

1) We have tested the role of the previously-reported GSK3 β sites on PD-L1 (T180 and S184)⁸ and included the results in Figure 3a shown below. Mutating the residues to the phosphorylation-resistant alanine showed that these sites also play an important role in ES-072-induced PD-L1 degradation.

2) In order to describe our observations more accurately, we have now changed our wording in the manuscript (also shown below).

"In agreement with the data which shows that Ser279/Ser283 phosphorylation is important for ES-072-induced PD-L1 degradation,"

We hope that these responses clarify the issue.

8. Li, C. W. et al. Glycosylation and stabilization of programmed death ligand-1 suppresses T-cell activity. *Nat. Commun.* 7, 12632 (2016).

#3. The authors proposed that insufficient depletion of GSK3a by the knockdown strategy and contribution of other EGFR-regulated kinases could explain that ES-072 induced PD-L1 degradation in GSK3a-knockdown cells was still substantial. Again, these possibilities need to be tested by experiments. If GSK3a turns out to be only one of the

players in PD-L1 degradation, their major conclusion will need to be revised - ts. If GSK3a turns out to be only one of the pf PD-L1 degradation and cancer escape from immunity via EGFR-GSK3-ARIH1 signaling and identify GSK3 α and ARIH1 might be potential therapeutic targets to boost anti-tumor immunity and enhance immune checkpoint blockade immunotherapies.

Response:

We thank the reviewer for this comment.

1) In addition to GSK3 α , there are other kinases (e.g., GSK3 β ⁸) that promote PD-L1 degradation downstream EGFR inhibition.

2) In order to describe our observations more accurately, we have now changed our wording in the manuscript shown below.

“Knockdown of GSK3 α in U937 and H1975 cells partially rescued PD-L1 degradation induced by ES-072 (Fig. 3i and Supplementary Fig. 4g).”

“Our results delineate a mechanism of PD-L1 degradation and cancer escape from immunity via EGFR-GSK3 α -ARIH1 signaling and suggest GSK3 α and ARIH1 might be potential drug targets to boost anti-tumor immunity and enhance immunotherapies.”

We hope that these responses clarify the issue.

8. Li, C. W. et al. Glycosylation and stabilization of programmed death ligand-1 suppresses T-cell activity. *Nat. Commun.* **7**, 12632 (2016).